# TempestExtremes v2.1: A Community Framework for Feature Detection, Tracking and Analysis in Large Datasets

Paul A. Ullrich[1], Colin M. Zarzycki[2], Elizabeth E. McClenny[1], Marielle C. Pinheiro[1], Alyssa M. Stansfield[3], and Kevin A. Reed[3]

[1]Department of Land, Air and Water Resources, University of California, Davis, Davis, California
[2]Pennsylvania State University
[3]School of Marine and Atmospheric Sciences, State University of New York at Stony Brook, Stony Brook, New York

**Correspondence:** Paul Ullrich (paullrich@ucdavis.edu)

**Abstract.** TempestExtremes (TE) is a multifaceted framework for feature detection, tracking, and scientific analysis of regional or global Earth system datasets on either rectilinear and unstructured/native grids. Version 2.1 of the TE framework now provides extensive support for examining both nodal (i.e., pointwise) and areal features, including tropical and extratropical cyclones, monsoonal lows and depressions, atmospheric rivers, atmospheric blocking, precipitation clusters, and heat waves. Available operations include nodal and areal thresholding, calculations of quantities related to nodal features such as accumulated cyclone energy and azimuthal wind profiles, filtering data based on the characteristics of nodal features, and stereographic compositing. This paper describes the core algorithms (kernels) that have been added to the TE framework since version 1.0, including algorithms for editing pointwise trajectory files, composition of fields around nodal features, generation of areal masks via thresholding and nodal features, and tracking of areal features in time. Several examples are provided of how these kernels can be combined to produce composite algorithms for evaluating and understanding common atmospheric features and their underlying processes. These examples include analyzing the fraction of precipitation from tropical cyclones, compositing meteorological fields around extratropical cyclones, calculating fractional contribution to poleward vapor transport from atmospheric rivers, and building a climatology of atmospheric blocks.

## 1 Introduction

For many atmospheric and oceanic features, automated object identification and tracking in large datasets has enabled targeted scientific exploration of feature-specific processes. Software tools for feature tracking, colloquially referred to as "trackers", are valuable for evaluating model performance (Davini and D'Andrea, 2016; Stansfield et al., 2020); understanding upstream process drivers, such as large-scale meteorological patterns (e.g. Grotjahn et al., 2016); and projecting future changes in feature characteristics and climatology (Roberts et al., 2020a). When well-engineered, these automated tools provide a means for analyzing the multiple petabytes of climate data now available and anticipated in the next decade (Schnase et al., 2016; Hassani et al., 2019). Since its introduction, TempestExtremes (TE, Ullrich and Zarzycki, 2017) has been continuously augmented with new kernels – that is, basic data operators that can act as building-blocks for more complicated tracking algorithms – designed to streamline data analysis and generalize capabilities present in other trackers. These kernels thus provide more options and

flexibility in exploring the space of trackers for each feature, and enable a deeper understanding of how robust a given scientific

conclusion is with respect to the choice of tracker. We describe the most significant of these updates and provide a number of use cases to demonstrate TE's functionality for real scientifically driven case studies.

Numerous publications over the past several decades have presented automated algorithms for identification of both nodal (i.e., pointwise) and areal atmospheric features. Ullrich and Zarzycki (2017) Appendices A-C summarized dozens of such automated algorithms for extratropical cyclones, tropical cyclones, and tropical easterly waves. Even so, work to identify

optimal tracking criteria continues (Murata et al., 2019). Beyond these traditionally tracked features, many recent papers have focused on defining regionally relevant features such as monsoonal lows and depressions, associated with heavy precipitation in monsoonal regions (Hurley and Boos, 2015; Vishnu et al., 2020). Areal feature tracking algorithms have also been developed for clouds (Heikenfeld et al., 2019), atmospheric rivers (Shields et al., 2018; Rutz et al., 2019), atmospheric blocking (Scherrer et al., 2006), mesoscale convective systems (Prein et al., 2017; Feng et al., 2018), precipitation clusters (Clark et al., 2014;

Pendergrass et al., 2016), convectively-generated outflow boundaries (Chipilski et al., 2018), gust fronts (Delanoy and Troxel, 1993), and frontal systems (Hope et al., 2014; Schemm et al., 2015; Parfitt et al., 2017). Both nodal and areal algorithms generally feature a similar set of kernels, motivating the development of a single package encompassing relevant capabilities. For example, the majority of detection algorithms are built upon an algorithmic paradigm known as MapReduce (Dean and Ghemawat, 2008), where individual time slices are assessed independent of one another (an embarrassingly parallel "map"

operation) then combined via a serial "reduce" operation. By building a single framework for distributing time slices to different feature identification algorithms, then combining multiple features into a single dataset, we can avoid duplication of this infrastructure across multiple trackers. Leveraging commonalities such as these enables improvements in algorithmic efficiency to be simultaneously administered to multiple trackers, and reduces redundancies from algorithmic validation and testing.

TE has been engineered with the goal of providing a comprehensive and user-friendly toolbox for feature tracking in model,

reanalysis, or observational data products. It features a set of core design principles to enable its easy application in scientific analyses:

- The TE kernels are encapsulated in a variety of executables that are fully configurable from the command line (i.e., containing no hard-coded thresholds). Thus the processing operations performed by TE can be easily conveyed simply by communicating the relevant command(s).

- TE abstracts many of the finer details about the structure of climate datasets through the use of physically-motivated kernels (such as the closed-contour operator), physically-based units, and internal indexing with Climate and Forecast compliant (CF-compliant) time variables.

- TE directly addresses the need for high-throughput, readily usable, and standardized data analysis tools. Its kernels are individually implemented in optimized and, where appropriate, parallelized C++.

- TE also addresses a growing need for data analysis tools that work with "big data", enabling significant data volume reduction by isolating characteristics of individual features rather than full fields.

– TE's algorithmic kernels are designed for arbitrarily grids, recognizing that climate models have largely moved away from latitude-longitude grids and towards quasi-uniform grids (Ullrich et al., 2017).

– TE is a fully open-source product, publicly developed and distributed via GitHub with permissive open source licensing.

These principles complement the underlying foci motivating TE's development: robustness, usability, maintainability, and extensibility. To the best of the authors' knowledge, no other comprehensive toolkit exists for general nodal and areal feature tracking in Earth system datasets.

The remainder of this paper follows an analogous structure to Ullrich and Zarzycki (2017): Section 2 describes the core algorithms and kernels now available in TE version 2.1. In section 3, we present several examples of how these kernels can 65 be combined together to form recipes for tracking tropical cyclones (TCs), for calculating fractional contribution of precipitation from TCs, for tracking and compositing extratropical cyclone fields, for tracking atmospheric rivers, and for tracking atmospheric blocks. A summary of results and future work is given in section 4.

## 2 TempestExtremes algorithms and kernels

In this section we describe the kernels available in the TE software package, organized by executable, with an emphasis on 70 additions since TE version 1.0. Technical details on the operation of TempestExtremes can be found in the user guide (Ullrich, 2020).

### 2.1 DetectNodes and StitchNodes

DetectNodes (formerly DetectCyclonesUnstructured) is used for the detection of nodal feature candidates, and corresponds to the parallel "map" step in the "MapReduce" framework – that is, candidate points are first selected based on information at 75 a single time slice. DetectNodes is typically followed by StitchNodes, which represents the serial "reduce" operation in the chain. StitchNodes connects nodal features together in time and produces paths associated with singular features. Both of these executables and their algorithmic kernels are described in Ullrich and Zarzycki (2017), although v2.1 now supports the use of physical time units for thresholds and time subsetting. For example, `mintime` may be specified as a minimum number of time slices (e.g., `"5"`) or as the minmum number of hours between first and last candidate in a path (e.g., `"24h"`).

DetectNodes and StitchNodes output trajectories in a format originally defined by the GFDL tropical cyclone tracker (TSTORMS; Vitart et al., 1997; Zhao et al., 2009). These files are generally referred to as *nodefiles*.

### 2.2 NodeFileEditor

NodeFileEditor is a new addition to TE for editing nodefiles. It includes options for (1) appending new details to trajectories, such as radial wind profiles or accumulated cyclone energy, (2) removing certain columns from nodefiles, or (3) filtering 85 trajectories or points along a trajectory, e.g., when outside of a specific time interval. A list of functions currently available in NodeFileEditor are given in Table 1. These functions may be chained to perform multiple related operations, such as computing

| Operator | Description |
|---|---|
| eval_ace | Calculate the instantaneous accumulated cyclone energy (ACE, Bell et al., 2000), equal to $10^{-4}u_{kt,max}^2$ where $u_{kt,max}$ is the maximum wind speed within a prescribed radius of the nodal feature, in knots. We use a value of 1.94384 kt $(\text{m s}^{-1})^{-1}$ to convert $\text{m s}^{-1}$ to kt. |
| eval_acepsl | Approximate ACE using sea level pressure to predict surface wind speed (ACEPSL). Currently, ACEPSL is calculated as ACE, but using $u_{kt,max} = 1.94384\,\text{kt}\,(\text{m s}^{-1})^{-1} \times 3.92 \times (1016.0\,\text{hPa} - psl_{min})^{0.644}$ (Holland, 2008), where $psl_{min}$ is the minimum sea level pressure within a prescribed radius. |
| eval_ike | Calculate the instantaneous integrated kinetic energy (Powell and Reinhold, 2007), defined as $\sum_i \frac{1}{2}u_i^2 A_i$, where $u_i$ is the magnitude of the wind speed at that grid cell (in $\text{m s}^{-1}$), $A_i$ is the area of that grid cell in $\text{m}^2$, and the sum is taken over all grid cells within a prescribed radius. |
| eval_pdi | Calculate the power dissipation index (Emanuel, 2005), defined as $u_{max}^3$, where $u_{max}$ is the maximum wind speed within a prescribed radius in $\text{m s}^{-1}$. |
| radial_profile | Develop a radial profile of the specified variable at each time slice around the nodal feature point by binning by radial distance and averaging gridpoint values. The output is expressed using python array syntax. |
| radial_wind_profile | As radial_profile but for the radial and azimuthal wind speed. The radial and azimuthal components are computed by projecting the 2D velocity at each grid point onto the radial and azimuthal vector fields around each nodal feature prior to binning. |
| lastwhere | Given an array as input, such as the output of radial_profile, identify the distance or index of the array that satisfies a given threshold. For example, this operator is used for determining the radius at which azimuthal wind speed is greater than 8 $\text{m s}^{-1}$. |
| value | Given an array, extract the value at the specified index using linear interpolation where needed. |
| max_closed_contour | For a given field, determine the largest value that satisfies the closed contour criteria (see Ullrich and Zarzycki (2017) section 2.6) around each nodal feature (see Algorithm 1). |
| region_name | Given the names and coordinates of polygons in longitude-latitude space, identify the name of the region for a given pointwise feature. Each point is identified as being in a given region using a straight-line test along lines of constant latitude. If the number of intersections with edges of the polygon is odd (even), then the point is inside (outside). |

**Table 1.** Functions implemented in NodeFileEditor as of TempestExtremes version 2.1.

a radial wind profile of a tropical cyclone and then extracting the radius where a particular wind threshold is exceeded. An example of such chaining of commands is given in section 3.3.

Most of the implemented algorithms are straightforward, except for `max_closed_contour`, whose pseudocode is provided in Algorithm 1. Intuitively, this algorithm can be thought of as filling up a 3D extruded surface representative of the contours of the field until fluid spills farther out than the prescribed maximum distance. The last height difference is then recorded as the maximum delta for the closed contour.

**Algorithm 1** Determine the maximum closed contour delta (largest field delta that permits a closed contour within the given distance of the feature) for each node in a given nodefile `N`, over field `F` and maximum distance `maxdist`. This algorithm uses a priority queue, which places the node with the highest priority (in this case the smallest delta) at the top of the queue.

```
max_delta = max_closed_contour(nodefile N, field F, maxdist)

  for each node n in N
    max_delta[n] = 0
    define empty priority_queue pqueue
    insert node n into pqueue with delta 0
    visited = []
    while pq is not empty
      p = remove node from pqueue with lowest delta
      add p to visited
      for all neighbors q of p
        if q is not in visited then add q to pqueue with delta (F[q] - F[n])
      if (dist(p,n) < maxdist) and (F[p] - F[n] > max_delta[n]) then
        max_delta[n] = F[p] - F[n]
```

## 2.3 NodeFileFilter

NodeFileFilter encapsulates algorithms for masking spatial data using nodefile information, i.e., effectively converting node-files into binary raster masks at each time slice and (optionally) applying them to available data. Filtering can be performed using the distance from each feature, based on the closed contour of each feature (as described by Algorithm 2), or by thresholding of areal regions that are within a given distance of each nodal feature (as described by Algorithm 3). The latter is useful for identifying, for instance, precipitation clusters associated with tropical cyclones.

## 2.4 NodeFileCompose

NodeFileCompose includes functionality for snapshotting fields around nodal features (i.e., at each time slice projecting fields onto the stereographic plane centered on a nodal feature) or compositing fields (i.e., averaging snapshots). In the same vein, it also includes functionality for snapshotting or compositing a particular geographic region when a feature is present. Stereographic composites are computed using Algorithm 4. The mathematical operators used for the local stereographic projection are given in Appendix A.

**Algorithm 2** Generate a binary mask using a closed contour criterion, given nodefile `N`, field `F`, closed contour magnitude `delta`, maximum mask distance `dist`, and maximum distance for minima/maxima search `minmaxdist`. The functions `find_min_near` and `find_max_near` are given by Algorithm 4 in Ullrich and Zarzycki (2017).

```
M = mask_by_closed_contour(nodefile N, field F, delta, dist, minmaxdist)

  for each node n in N
    if (delta > 0) m = find_min_near(n, F, minmaxdist)
    if (delta < 0) m = find_max_near(n, F, minmaxdist)
    visited = []
    tovisit = [m]
    ref_value = F[m]
    while tovisit is not empty
      p = remove node from tovisit
      if visited contains p then continue
      add p to visited
      if (dist(p,m) > dist) then continue
      if (sign(delta) * (F[p] - F[m]) > abs(delta)) then continue
      M[p] = 1
      add neighbors of p to tovisit
```

## 2.5 DetectBlobs

DetectBlobs is used for identifying areal features (blobs), such as atmospheric blocks, atmospheric rivers, or precipitation clusters. As with DetectNodes, this executable represents the parallel "map" step in the "MapReduce" framework. Candidate regions are selected based on information at a single time slice, typically simple thresholds such as "all points where precipitation is greater than 1 mm d$^{-1}$". Features are marked using a binary mask and output stored in NetCDF format. Contiguous regions may then be excluded based on either geometric thresholds or criteria derived from other variables. DetectBlobs supports MPI-based parallelism over input files.

## 2.6 StitchBlobs

StitchBlobs is used for tracking areal features (blobs) in time, assigning connected features a unique global id and/or applying time-dependent criteria to each contiguous region. Given input as a time-dependent binary mask variable, blobs that overlap at sequential time steps will be assigned the same global identifier. The algorithm implemented in TE for connecting blobs in

**Algorithm 3** Generate a binary mask by picking out blobs satisfying a threshold within a given radius of each node. The inputs include the nodefile N, field F, the search distance `searchdist`, threshold operation `threshold`, and maximum mask distance `maxdist`.

```
M = mask_by_nearbyblobs(nodefile N, field F, searchdist, threshold, maxdist)

  for each node n in N
    visited = []
    tovisit = [n]
    while tovisit is not empty
      p = remove node from tovisit
      if visited contains p then continue
      add p to visited
      if (dist(p,n) > searchdist) then continue
      add neighbors of p to tovisit

      if F[p] does not satisfy threshold then continue
      tovisitnested = [p]
      while tovisitnested is not empty
        q = remove node from tovisitnested
        if visited contains q then continue
        add q to tovisitnested
        if (dist(q,n) > maxdist) then continue
        if F[q] does not satisfy threshold then continue
        M[q] = 1
        add neighbors of q to tovisitnested
```

**Algorithm 4** Generate a stereographic composite of field `F` over the given set of nodes `N`, generated by DetectNodes. The stereographic grid has resolution `res` and grid spacing `dist`, given as the great circle distance along coordinate lines passing through the origin.

---

```
field C = stereographic_composite(field F, node_list N, res, dist)
  C = empty 2D stereographic grid with parameters (res,dist)
  for each node n in N
    G = generate 2D stereographic grid with parameters (res,dist) centered on n
    for each node m in G
      use inverse stereographic projection to obtain point p corresponding to G[m]
      q = nearest grid point in F to p
      assign G[m] to value F[q]
    C = C + G
  C = C / size(N)
```

---

time uses a forward-backward search that can treat the 2D space + 1D time object as a single object, allowing for both splitting and merging of features in time.

The pseudocode for this search protocol is provided in Algorithms 5 and 6, and its operation is illustrated in Figure 1. Put briefly, contiguous regions at each time slice are identified using a flood fill algorithm and assigned a unique tag of the form (time id, blob id). An additional "merge distance" argument can be specified that merges nearby blobs at each time slice if their perimeters are within this specified distance. A graph is then constructed with each of these tags corresponding to the nodes of the graph. Edges are then added to the graph where a pair of areal features are deemed to be connected in time. Since multiple edges could be generated to or from a feature on a given time slice, multiple mergers or splits may occur simultaneously. Finally, the components of the graph are each assigned a unique global id, with lower global ids corresponding to blobs that first appear at earlier times. In Figure 1, feature 1 and 2 at time index 1, denoted (1,1) and (1,2), will both be assigned the same global id since they are connected at a later time. Similarly, feature 1 and 2 at time index 3, denoted (3,1) and (3,2), are assigned the same global id since they were connected at an earlier time. Note that global ids start at 1 and are consecutive thereafter; they are assigned only after connected components of the graph are identified, and as such are unrelated to the blob id on each time slice.

By default, areal features are deemed to be connected in time if they share at least one grid point at subsequent time steps (regardless of the area of that grid point). For example, in Figure 1, areal regions (1,1) and (2,1) overlap in space and so are deemed to be connected. If a stricter threshold on the overlap area is needed for blobs at sequential time slices to be deemed part of the same cluster, StitchBlobs provides arguments for minimum overlap between the current blob and blobs at the previous

**Algorithm 5** Flood fill blobs on unstructured grid using breadth-first graph search over binary field G, merging blobs within the given distance merge_dist.

```
blob set array S =
  flood_fill_and_tag_with_merging(binary_field G, merge_dist)

  % Perform flood fill and tag
  current_tag = 0
  visited = []
  kdP = empty array of kdtrees
  P = array storing perimeter points for each tag
  for each node p
    if (F[p] is not 0) and (visited does not contain p) then
      current_tag = current_tag + 1
      tovisit = [p]
      while tovisit is not empty
        q = remove node from tovisit
        if visited contains q then continue
        add q to visited
        if (F[q] is not zero) then
          insert q into S[current_tag]
          add neighbors of q to tovisit
          if any neighbors r of q have (F[r] = 0) then
            add q to P[current_tag] and kdP[current_tag]

  % Build list of blobs to merge by distance
  M = empty graph with integer nodes denoting merged blobs
  for all ordered pairs of tags (s,t)
    for all nodes p in P[s]
      if nearest neighbor from kdP[t] is closer than merge_dist then
        add edge (s,t) to M

  % Merge blob sets
  for each tag t from 1 to current_tag
    find minimum tag s of connected subgraph containing t
    merge S[t] into S[s]
```

**Algorithm 6** Forward-backward algorithm for areal-feature search. Given a time series of binary fields `G[t]`, a merge distance `merge_dist` and overlap thresholds `min_overlap_*` and `max_overlap_*`.

```
field F[t] = stitch_blobs(binary_field G[t], merge_dist,
  min_overlap_prev, max_overlap_prev,
  min_overlap_next, max_overlap_next)

  % Use flood fill and merge to identify blobs
  for all times t from 0 to length(G)
    S[t] = flood_fill_and_tag_with_merging(G[t], merge_dist)

  % Build overlap graph
  M = empty graph denoting (time,blob) pairs
  for all times t from 0 to length(G)
    for all blobs p in S[t]
      insert node (t,p) into M

  % Identify blobs to be stitched together in time
  for all times t from 0 to length(G)-1
    for all blobs p in S[t]
      for all blob n in S[t+1]
        prev_area = area of S[t][p]
        next_area = area of S[t+1][n]
        overlap_area = overlap area between S[t][p] and S[t+1][n]
        if (overlap_area / prev_area >= min_overlap_prev) then
          and (overlap_area / prev_area <= max_overlap_prev)
          and (overlap_area / next_area >= min_overlap_next)
          and (overlap_area / next_area <= max_overlap_next)
            add edge ((t,p),(t+1,n)) to M
    S_prev = S_next

  % Assign a common global_id to overlapping blobs
  global_id = 1
  for all nodes (t,p) in M
    for each node (tx,px) in connected subgraph of M containing (t,p)
      for each node p in S[tx][px]
        F[t][p] = global_id
    global_id = global_id + 1
```

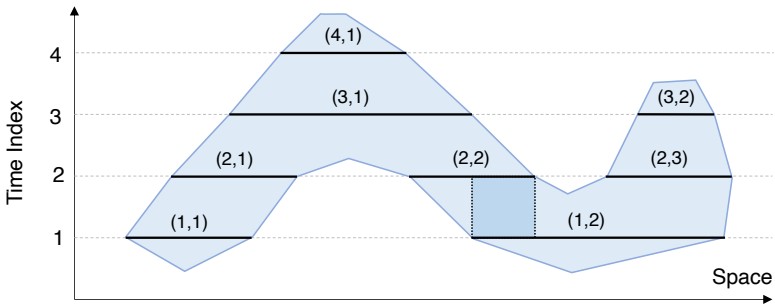

**Figure 1.** A depiction of Algorithms 5 and 6 for forward-backward areal feature search used by StitchBlobs, simplified to show one space dimension (e.g., longitude) and the time dimension.

and/or next timestep. In this example, blob tag (2,2) overlaps only 25% of the area of blob tag (1,2), meaning that (2,2) and (1,2) are deemed unconnected if the "minimum previous overlap" is greater than 25%. On the other hand blob tag (1,2) covers 50% of the area of blob tag (2,2), so these two would be deemed unconnected if the "minimum next overlap" is greater than 50%.

## 2.7    Other Utilities

In addition to the core functionality described in previous sections, TE also provides a number of other utilities to manage nodefiles, binary masks, and other climatological data relevant to feature tracking. These are briefly mentioned here as this functionality is employed in the composite tracking algorithms and analysis of section 3.

– *Climatology* is used for constructing climatological time series, including long-term daily, monthly, seasonal, and annual means. It supports parallel execution over files via MPI, as well as arguments that can be used to limit the amount of memory used by each thread.

– *FourierFilter* is used for Fourier filtering/smoothing of input data series. Although it provides a general implementation that could be used for any dataset, it has primarily been used for smoothing long-term daily means produced from Climatology.

– *VariableProcessor* provides direct access to TE's internal variable processing capability, allowing arithmetic and grid-based operations to be applied to gridded data files. The operation of this utility is roughly analogous to that of the NetCDF Operator `ncap2` (NCO; Zender, 2008).

## 3    Selected examples

In this section we present selected examples of tracking and analysis of different features; that is, different recipes for combining the algorithmic kernels described in section 2 to produce composite tracking and analysis algorithms. In all examples, the

corresponding TE commands are provided to both demonstrate that they are effective at conveying the operation in a human-readable manner and to enable reproducability of our results. Past examples from the literature using TE are provided in section 3.1. This is then followed by several examples from TE of feature-based tracking and subsequent analysis. These examples include TC tracking in ERA5, fractional contribution of precipitation from TCs in ERA5 and TRMM, atmospheric river tracking in ERA5, extratropical cyclone tracking in CMIP6 data, and finally generation of an atmospheric blocking climatology using MERRA2 data.

## 3.1 Examples from the existing literature

Since version 1.0, TE has been employed for feature tracking in a number of scientific studies. Here we catalogue known publications emerging from those studies, organized by feature type.

**Tropical Cyclones (TCs):** More than any other feature, TE has been employed for the study of TCs. TE was first employed as a TC tracker to understand intensity errors associated with one-way coupling between ocean and atmosphere in Zarzycki (2016). It was subsequently used to investigate the TC wind-pressure relationship in Chavas et al. (2017), a relationship later revisited in Moon et al. (2020) where TE was used to assess its sensitivity to model resolution. In Wing et al. (2019), TE was applied to native grid data produced using the Community Atmosphere Model Spectral Element (CAM-SE) dynamical core to track TCs; outputs were then used to investigate the processes underlying moist intensification of TCs. A related study by Camargo et al. (2020) used this dataset to investigate the large-scale environment around TCs. In Roberts et al. (2020a) and Roberts et al. (2020b), TE-derived TC tracks were used to understand resolution sensitivity and future change in both historical and future HighResMIP experiments (Haarsma et al., 2016) across several models. Along these lines, Balaguru et al. (2020) used TE to characterize TC climatology in the Energy Exascale Earth System Model (E3SM). Reed et al. (2020, 2021) used TE to extract tracks of Hurricanes Florence (2018) and Dorian (2019) and attribute human influence on these storms. TE has also been used for tracking storms in aquaplanet simulations (Chavas and Reed, 2019) so as to better understand how dynamic forcing impacts TC genesis and size. Recent work by Stansfield et al. (2020) has also leveraged some of the more advanced capabilities in TE to filter fields (e.g., precipitation) in the vicinity of tracked features to evaluate model performance. TE has also been used for tracking of TCs in extremely high-resolution regional simulations (Steptoe et al., 2021) and investigating TCs in paleoclimate simulations (Kiehl et al., 2021).

**Extratropical Cyclones (ETCs):** In order to better understand cyclonic storms and their impacts, Zarzycki et al. (2017) developed the ExTraTrack software framework atop TE to track TCs and ETCs through their entire lifecycle. This module enabled cyclonic storms to be examined using the thermal wind and thermal asymmetry phase space of Hart (2003). ExTraTrack was later applied to a suite of high-resolution global simulations in Michaelis and Lackmann (2019) and Michaelis and Lackmann (2021). ETCs were also tracked in the Community Earth System Model Large Ensemble (CESM-LENS) in Zarzycki (2018) to understand the drivers responsible for snowstorms in the US Northeast. Then in Small et al. (2019), extratopical storms tracked using TE were used to determine if resolving ocean fronts improves the representation of simulated storm tracks. In Zhang et al. (2021) the vertical symmetry criterion from ExTraTrack was also adapted for tracking of Mediterranean Hurri-

canes (Medicanes). Finally, TE was also used to track ETCs as part of an effort to evaluate severe local storm environments in climate models and reanalysis (Li et al., 2020).

**Monsoonal lows and depressions:** Analogous to the study of Zarzycki and Ullrich (2017), Vishnu et al. (2020) optimized DetectNodes for tracking of monsoon lows and depressions. A comprehensive analysis of input fields found that 850hPa streamfunction tended to produce better results compared with trackers based on sea level pressure, vorticity, and geopotential. A weighted Critical Success Index (CSI) (Di Luca et al., 2015) was used to determine tracker performance. However, acknowledging the possibility of errors in the reference dataset (here the Sikka archive), the weighted CSI index used in this analysis also considered the degree to which a track is represented similarly across all reanalyses. A related study by Zhang et al. (2019) also tracked tropical depressions in the North Indian Ocean in 2018 to investigate anthropogenic impact on this storm season, and a recent study by You and Ting (2021) used TE to assess trends in South Asian Monsoon low pressure systems.

**Atmospheric blocking:** In Pinheiro et al. (2019), a suite of atmospheric blocking methods from TE were applied to ERA-Interim data to better understand sensitivities of atmospheric blocks to the detection algorithm and the meteorological environment around blocking features.

**Atmospheric rivers (ARs):** Atmospheric river tracking with TE was first documented as part of the Atmospheric River Transport Method Intercomparison Project (ARTMIP) in Shields et al. (2018), and later in Rutz et al. (2019). The proposed algorithm used the Laplacian of the integrated vapor transport (IVT) field rather than the IVT field itself, thus flagging IVT "ridges" rather than IVT over a threshold; this choice addressed issues of stationarity generally present in trackers using an IVT threshold. TE's algorithm has since been used both for AR detection and tracking (with DetectBlobs and StitchBlobs) in several subsequent studies (Rhoades et al., 2020b, a; Patricola et al., 2020; McClenny et al., 2020; Huang et al., 2021; Zhou et al., 2021).

## 3.2 Tropical Cyclone Tracking in ERA5

In Zarzycki and Ullrich (2017), a sensitivity analysis was applied to optimize TE for the detection of tropical cyclones by benchmarking hit rate (HR) and false alarm rate (FAR) from reanalysis products against the International Best Track Archive for Climate Stewardship (IBTrACS; Knapp et al., 2010). The resulting configuration, which tracked storms based on sea level pressure minimum, produced the highest HR minus FAR differential in the literature across a wide range of reanalysis products. An interesting result that emerged from this analysis was that upper level geopotential layer thickness (typically Z300 minus Z500) was the most robust indicator of an upper level warm core across products. In this section we apply the same configuration that provided maximal agreement between earlier-generation reanalyses and IBTrACS to ERA5 input (Hersbach et al., 2020) so as to identify ERA5 TC tracks.

### 3.2.1 Step 1: Identify candidate storms

Tropical cyclone tracking is an exemplar of the MapReduce paradigm discussed earlier in this paper. Essentially all published algorithms for TC tracking (e.g., Ullrich and Zarzycki, 2017, Appendix B) make use of a two-step process consisting of first detecting TC candidates, then stitching together candidates in time. Both steps of this process include hard thresholds that

separate TCs from related features. Although TE allows users to vary the values of these thresholds, here we only consider one such variation of these parameters.

    In the TC detection algorithm described in Zarzycki and Ullrich (2017), candidates are defined as are points that have both a sea-level pressure minimum and an upper level warm core. These conditions are codified via the command:

```
DetectNodes --in_data_list ERA5_TC_files.txt --timefilter "6hr" \
   --out_file_list ERA5_DN_files.txt \
   --searchbymin MSL \
   --closedcontourcmd "MSL,200.0,5.5,0;_DIFF(Z(300hPa),Z(500hPa)),-58.8,6.5,1.0" \
   --mergedist 6.0 \
   --outputcmd "MSL,min,0;_VECMAG(VAR_10U,VAR_10V),max,2;ZS,min,0"
```

For this example, our ERA5 data comes from the NCAR Research Data Archive (European Centre for Medium-Range Weather Forecasts, 2019), with 3D time-series data provided at hourly resolution in daily chunks, 2D time-series data provided at hourly resolution in monthly chunks, and 2D invariant data provided in a single file. The `timefilter` argument here indicates that data should be downselected to six-hourly, which is typical for analysis of TCs. As different variables are distributed across multiple files, the first two lines of the input data consist of several files containing 3D geopotential height on pressure

surfaces (Z), 2D mean sea level pressure (MSL), 2D 10 meter zonal and meridional wind speeds (VAR_10U and VAR_10V), and surface elevation (ZS), separated by semicolons. Note that TE supports different agglomerations of time slices, as it uses the CF-compliant time to match time slices across files.

    To first limit the search space of possible TCs, we identify candidates as local minima in the sea level pressure field. Two closed contour criteria are used to eliminate candidates. As argued in Ullrich and Zarzycki (2017), closed contour criteria are a

240 more physically grounded way of defining features since they can be employed for both discrete and continuous fields – as opposed to, e.g., "gridpoint maxima" that are inherently sensitive to the dataset's grid structure and spacing. The first criterion we use for TCs is `"MSL,200.0,5.5,0"`, which indicates that mean sea level pressure must increase by 200 (Pa) over a distance of $5.5°$ great-circle-distance (GCD) from the candidate point (the low pressure region must be of sufficient magnitude and sufficiently compact to be considered coherent). The second criterion is `"_DIFF(Z(300hPa),Z(500hPa)),-58.8,6.5,1.0"`

which indicates that the difference between geopotential (Z) on the 300hPa and 500hPa surfaces must decrease by 58.8 m$^2$ s$^{-2}$ (equal to 6 m geopotential height) over a distance of $6.5°$ GCD, using the maximum value of this field within $1°$ GCD as reference. This second criterion indicates that there must be a coherent upper-level warm core attached to the local low so as to structurally differentiate these features from extratropical systems. It is also an example of TE's ability to evaluate functions of meteorological fields at run-time. Finally, candidates that have been identified with this protocol are eliminated if a stronger

minimum exists within 6 degrees great circle distance.

    The remaining `outputcmd` argument indicates three additional outputs that are calculated and written as additional columns in each nodefile. Here `"MSL,min,0"` outputs the value of MSL at the candidate point, `"_VECMAG(VAR_10U,VAR_10V),max,2"` outputs the maximum magnitude of the vector wind at 10 m altitude within $2°$ GCD of the candidate, and `"ZS,min,0"` out-

puts the surface height at the candidate point. These variables are needed in the subsequent StitchNodes step to construct and
filter TC trajectories.

### 3.2.2 Step 2: Connect candidate storms together in time

Once TC candidates have been identified on each time slice, the "stitching" step in the algorithm ties these candidates together in time to form TC trajectories (the "Reduce" operation in the MapReduce paradigm). Some features that are too weak, too short-lived, or too disorganized are eliminated from contention at this stage. Also, features that are more likely related to topographic anomalies rather than real storms are also removed.

To build these trajectories with TE, we apply the StitchNodes command to the output from Step 1:

```
StitchNodes  --in_list ERA5_DN_files.txt --out ERA5_TC_tracks.txt \
  --in_fmt "lon,lat,slp,wind,zs" \
  --range 8.0 --mintime "54h" --maxgap "24h" \
  --threshold "wind,>=,10.0,10;lat,<=,50.0,10;lat,>=,-50.0,10;zs,<=,150.0,10"
```

The first three arguments here indicate the input candidate nodefile (produced by DetectNodes) and the output nodefile. The format of these files differ because they convey different information – the former containing candidates detected at each time slice, and the latter containing paths, or lists of candidates from different time slices. Nonetheless auxiliary candidate information computed with DetectNodes' `outputcmd` is preserved.

The relevant tuning parameters are specified by `range`, `mintime` and `maxgap` and refer to the maximum distance (in degrees GCD) that a feature can move between subsequent detections, the minimum persistence time of each trajectory (calculated as the time between initiation and termination), and the maximum duration between two sequential detections, respectively. In particular, `maxgap` is a novel option that allows for a path to be missing candidates for some time slices (for instance due to temporary weakening of TC as it passes over land).

Four field-dependent thresholds are then specified for a trajectory to be accepted. The first threshold `"wind,>=,10.0,10"` indicates that the wind magnitude (derived from the "wind" column in the nodefile) must be greater than 10 m s$^{-1}$ for at least 10 time slices; this ensures that these features are sufficiently intense to be classified as tropical storms. The next two thresholds `"lat,<=,50.0,10;lat,>=,-50.0,10"` indicate that the latitude of the feature must be between 50S and 50N for at least 10 time slices, so as to eliminate any extratropical features that could not have existed as tropical storms. The final threshold `"zs,<=,150.0,10"` indicates that the feature must exist at an elevation below 150 m for at least 10 time slices; this removes false alarms that can often appear in regions of rough topography associated with PSL correction formulations.

### 3.2.3 Results from the generation of tropical cyclone trajectories

Figure 2 depicts the tropical cyclone trajectories produced from this analysis in ERA5, along with IBTrACS over the same period for reference. Storms are color-coded by sea level pressure, as opposed to surface winds, since it has been found that the former is better resolved in coarser datasets (Chavas et al., 2017). With that said, this procedure may overestimate storm

intensity at higher latitudes where storms are beginning to undergo extratropical transition. While ERA5 tracked storms are generally too weak in aggregate (lower density of orange and red trajectories in top panel), a common problem amongst reanalyses (Schenkel and Hart, 2012; Murakami, 2014; Hodges et al., 2017), the method shows high spatial and temporal correlation of storm climatology when compared to pointwise observations, and produces superior hit rates (78% for all TCs and 95% for those with wind speeds exceeding 33 m s$^{-1}$) and false alarm ratios (14% globally) when compared to many legacy tracking techniques (Zarzycki et al., 2021).

The TC detector described in this section was run on the NERSC Cori supercomputer on one node and using 32 threads. When run over the ERA5 data from January 1979 through February 2020 at 6 hourly temporal resolution, with 15,035 daily files, DetectNodes required 140 minutes run time. DetectNodes on Cori is strongly I/O bound, with reads from NetCDF files responsible for 81% of the total runtime. StitchNodes required 4 minutes and 55 seconds to process all 15,035 outputs from DetectNodes.

### 3.3 Fractional contribution of precipitation from TCs

For the examples here, we calculate the fractional contribution of precipitation from TCs for one reanalysis dataset, ERA5, and one observational dataset, the Tropical Rainfall Measuring Mission (TRMM3B42; Huffman et al., 2007). For ERA5, the TC track files are created as described in Section 3.2. For TRMM, the IBTrACS dataset is used for TC track observations. Because there are limited comprehensive and long-term observational datasets of complete TC wind fields, ERA5 wind field data is combined with IBTrACS to calculate the outer radius, r8, at every timestep for all historical TC tracks for the TRMM analysis.

#### 3.3.1 Step 1: Compute the outer radius of each tracked TC

As argued by Schenkel et al. (2017), the largest radius outside of the eyewall where the azimuthally-averaged wind speed exceeds 8 m s$^{-1}$ (r8) tends to be a good measure of the outer size of a TC. In Stansfield et al. (2020), TE was used to examine the distribution of r8 among TCs in reanalysis data in ERA5 and a series of runs from the Community Earth System Model (CESM). This paper further compared and contrasted TC-related precipitation within r8 against precipitation within a fixed distance of 500 km. We thus follow Stansfield et al. (2020) and use r8 as our criterion for grid points to be part of the TC.

To begin, radial profiles and radius of 8 m s$^{-1}$ wind are added to the nodefile generated in section 3.2.2 and the IBTrACS nodefile (not shown):

```
NodeFileEditor --in_data_list ERA5_TC_files.txt --in_nodefile ERA5_TC_tracks.txt \
  --in_fmt "lon,lat,slp,wind,zs" --out_nodefile ERA5_TC_radprofs.txt \
  --out_fmt "lon,lat,rsize,rprof" --time_filter "6hr" \
  --calculate "rprof=radial_wind_profile(VAR_10U,VAR_10V,159,0.125); \
  rsize=lastwhere(rprof,>,8)"
```

The input to this operation includes the files containing the 2D ERA5 10 meter zonal and meridional wind speeds (VAR_10U and VAR_10V), and the nodefile generated in section 3.2.2. As part of this analysis we also augment an IBTrACS nodefile

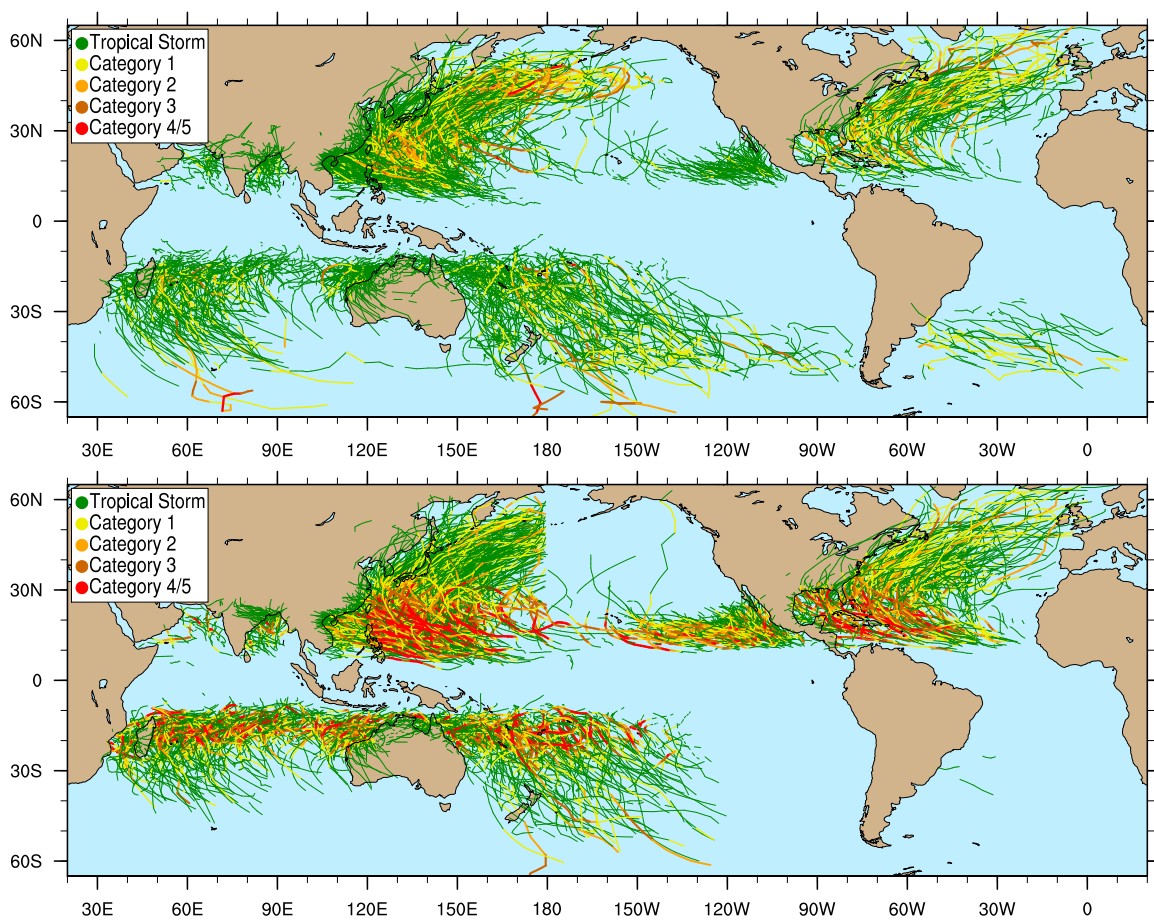

**Figure 2.** Tropical cyclone trajectories from ERA5 (top) and IBTrACS (bottom) over the period 1980-2019, inclusive. TempestExtremes is used to track TCs in ERA5, while pointwise observation are used for IBTrACS. Coloring denotes the instantaneous Saffir-Simpson category of the tropical cyclone. Category is computed from sea level pressure and applying the pressure-wind relationship of Atkinson and Holliday (1977) with updated coefficients from Knaff and Zehr (2007). The discontinuity at 180° longitude in the bottom panel is due to historical forecast center responsibilities.

in a similar manner, using the IBTrACS TC tracks, but ERA5 winds to estimate TC size (command not shown here). As in section 3.2.1, a time filter is used to only analyze 6-hourly time slices of data. Internal to the execution of this command is the construction of a date object for each entry of the nodefile, which is then cross-referenced against every time slice in the list of datafiles to find the corresponding field – in this way indexing is abstracted from the user.

The calculations requested from NodeFileEditor are specified by the `calculate` argument, executed from left to right. First the radial profile is computed with `radial_wind_profile(VAR_10U,VAR_10V,159,0.125)` and stored in variable `rprof`. These arguments indicate which variables should be used for the calculation, and that the radial profile should consist of 159 bins of width 0.125 degrees GCD. After the radial profile is calculated, the last value where the radial wind profile is greater than 8 m s$^{-1}$ is located and output to the nodefile. The last value in the array is taken because we want to avoid recording the radius of the 8 m s$^{-1}$ wind within the TC inner core. The number of bins and bin width were chosen based on the horizontal grid spacing of the ERA5 wind data, which is approximately 31 km. The bin width of 0.125$^\circ$ was chosen to adequately sample points at this grid spacing to create the radial wind profiles. The number of bins was chosen to ensure the radial averaging extended out far enough from the TC center points to capture the storms' complete wind circulations.

### 3.3.2   Step 2: Build a mask using the outer radius of storm

With the r8 value for each TC now in hand, we define "TC-related precipitation" as any precipitation which occurs in grid points that are considered part of a TC. Employing TE's NodeFileFilter command, precipitation outside of the circle with radius r8 centered on each TC is set to zero:

```
NodeFileFilter --in_nodefile ERA5_TC_radprofs.txt --in_fmt "lon,lat,rsize,rprof" \
   --in_data_list ERA5_precip_files.txt \
   --out_data_list ERA5_filtered_precip_files.txt \
   --var "PRECT" --bydist "rsize"
```

The input nodefile is the output nodefile from NodeFileEditor, now augmented with the radius of 8 m s$^{-1}$ winds. The input data contain 6-hourly ERA5 precipitation data from the NCAR Research Data Archive (European Centre for Medium-Range Weather Forecasts, 2019) under variable name `PRECT`. Precipitation in ERA5 is calculated from hourly forecasts initialized from the analysis at 06:00UTC and 18:00UTC. Precipitation is converted from hourly to 6-hourly by adding up the accumulated precipitation 3 hours before and 3 hours after the desired timesteps of 00:00UTC, 06:00UTC, 12:00UTC, and 18:00UTC. For example, to calculate 6-hourly precipitation at 06:00UTC, the precipitation from 03:00UTC to 09:00UTC is added up. For the TRMM analysis, the TRMM precipitation data is originally 3-hourly, so before analysis the TRMM data is summed into 6-hourly data. This is done using a centered averaging method, so for example, to calculate 6-hourly precipitation at 06:00UTC, half of the 03:00UTC precipitation, all of the 06:00UTC precipitation, and half of the 09:00UTC precipitation are added up. Output consists of a sequence of NetCDF files, one for each input file, containing filtered precipitation.

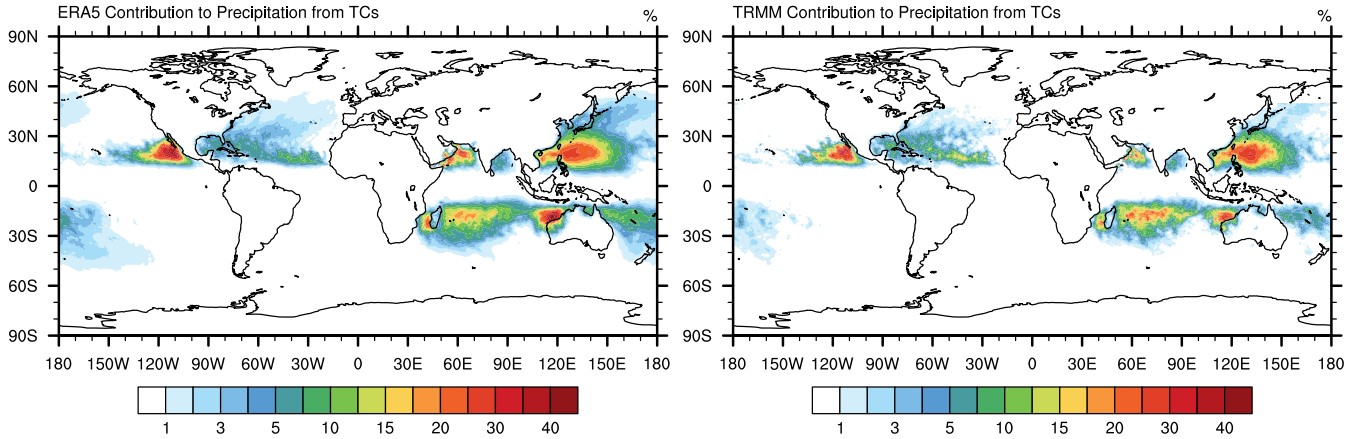

**Figure 3.** Percent contribution to precipitation from tropical cyclones using precipitation field from (left) ERA5 and (right) TRMM.

The final argument specifies how the filtering is performed, in this case "by distance" using `rsize`. This procedure only keeps precipitation grid point values that are within this distance of each detected TC. Internally to NodeFileFilter, the mask is computed through the employ of a kd-tree (see discussion in Ullrich and Zarzycki, 2017).

### 3.3.3 Results from the generated tropical cyclone precipitation climatology

Figure 3 shows the percent contribution to global precipitation from TCs for ERA5 and TRMM, calculated by using NCO's `ncra` to sum up the TC precipitation within r8 filtered by NodeFileFilter and dividing it by the sum of the total precipitation over the entire length of the datasets (1985-2019 for ERA5 and 1998-2014 for TRMM). The areas of largest TC contribution align with the areas of the highest TC activity shown in Figure 2 and typically occur over the ocean, in broad agreement with Prat and Nelson (2013). Khouakhi et al. (2017) (their Figure 3b) made a similar plot, except using land-based gauge data and for a slightly different time period, and showed similar locations of maximum contributions of 40-50% over northwestern Australia and eastern Asia.

### 3.4 Extratropical cyclones

Extratropical cyclones (ETCs) are mid-latitude, synoptic scale weather features responsible for a host of impacts, including high winds, coastal surge, and heavy precipitation, which can fall as rain, snow, sleet, or freezing rain (Schultz et al., 2019; Dacre, 2020). Even though these features occur at relatively large spatial scales, models still have difficulty in capturing hazards related to ETCs (e.g., Colle et al., 2015; Catalano et al., 2019), emphasizing the importance of evaluating them at a process level in weather and climate datasets.

Here we produce two-dimensional composites of several field associated with ETCs tracked in the first historical member of the Community Earth System Large Ensemble (Kay et al., 2015). ETCs over the northeastern United States were originally analyzed in this dataset in Zarzycki (2018). The years available for analysis in the historical simulations range from 1990-2015,

inclusive. We also apply a pre-defined intensity threshold and spatially constrain ETCs to pass over the continental United States (CONUS) in order to demonstrate a regional analysis and highlight both the filtering and compositing capabilities of TE.

### 3.4.1 Step 1: Generate extratropical cyclone trajectories

The algorithm applied here identifies cyclonic storms as sea level pressure minima. To avoid topographic lows and other features that are not meteorological in character, additional criteria are imposed on the minimum lifetime and propagation distance. Note that while the algorithm is highly similar to that published in Zarzycki (2018), other ETC detection algorithms analogous to those published in the Intercomparison of Mid Latitude Storm Diagnostics (IMILAST; Neu et al., 2013) can be configured using TE's command line options. These alternative approaches include tracking on low-level geopotential height (vorticity) minima (maxima), filtering based on spatial gradients, and removing candidate storms over higher terrain (see Table 1 in Neu et al. (2013)). Also, while a more complex algorithm could help eliminate cyclones that are tropical in nature (e.g., by using the `no_closed_contour` argument to eliminate candidates with an upper level warm core), one is not applied here due to the relatively low resolution of CESM LENS. These coarser grid spacings are generally insufficient to resolve TCs (Walsh et al., 2015), although higher resolution evaluations of ETCs may require additional exclusionary thresholds to minimize their inclusion in storm track datasets if desired.

To begin, cyclonic storms are identified using DetectNodes by following sea level pressure (here, PSL) minima in 6-hourly data:

```
DetectNodes --in_data_list B20TRC5CNBDRD.001.PS_list.txt --out cyclone_candidates \
    --closedcontourcmd "PSL,200.0,6.0,0" --mergedist 6.0 \
    --searchbymin "PSL" --outputcmd "PSL,min,0"
```

Our criterion for cyclonic storms is that the minimum pressure must be enclosed by a closed contour of 200 Pa within 6.0° of cyclone center. This minimum pressure location also defines the cyclone center. Candidates within 6.0° of one another are merged, with the lower pressure taking precedence. Outputs from DetectNodes are then concatenated into a single candidate list, and StitchNodes is run to track these features in time:

```
StitchNodes --in_fmt "lon,lat,slp" --in_list candidate_list.txt \
    --out etc-all-traj.txt \
    --range 6.0 --mintime 60h --maxgap 18h --min_endpoint_dist 12.0 \
    --threshold "lat,>,24,1;lon,>,234,1;lat,<,52,1;lon,<,294,1"
```

Here the StitchNodes thresholds require that storms persist for at least 60 hours, with a maximum gap (time between sequential detections satisfying the DetectNodes criteria) of at most 18 hours. Further, at least one point must pass through a geographic region (representing CONUS) bounded by 24°N and 52°N latitude and 234°E and 294°E longitude. We also require ETCs move at least 12° GCD from the start to the end of the trajectory, as specified by the `min_endpoint_dist` argument, in order to eliminate stationary features (e.g., the Icelandic Low) and spurious shallow lows generated over regions of high topography.

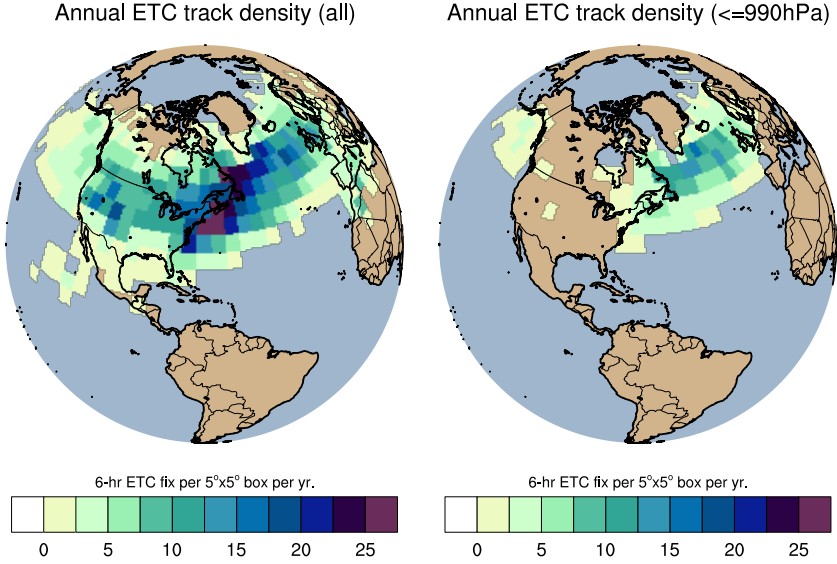

**Figure 4.** Track density maps for all CONUS ETCs tracked in the first historical member of CESM LENS (left) and only ETCs with simulated SLP less than or equal to 990 hPa (right). Units are number of six-hourly ETC occurrences per $5°$x$5°$ grid box per year.

### 3.4.2   Step 2: Filter out ETCs with sea level pressure above 990hPa

In some cases, it may be desirable to filter out the more intense, potentially more impactful, events. While the definition of a strong ETC is inherently subjective, we define a central pressure of 990 hPa as the demarcation between strong and moderate/weak ETCs as in Zhang and Colle (2017). To do so, all ETCs tracked in Step 1 are passed into NodeFileEditor, where a new trajectory file specified by argument `out_nodefile` is generated with storms only possessing intensities of 990 hPa or lower:

```
NodeFileEditor --in_nodefile etc-all-traj.txt  \
   --in_data_list B20TRC5CNBDRD.001.PRECT_list.txt \
   --in_fmt "lon,lat,slp" --out_fmt "lon,lat,slp" \
   --out_nodefile etc-strong-traj.txt \
   --colfilter "slp,<=,99000."
```

Figure 4 shows the annual track density of all ETCs tracked in Step 1 (left) and the same plot but with only the subset of ETCs stronger than 990 hPa included (right). These results broadly match those of other ETC trackers depicted in Fig. 1 of Neu et al. (2013), with a storm track belt extending across the North Atlantic centered on approximately 40-60°N latitude.

### 3.4.3 Step 3: Filter data within 25 degrees of storm center and composite

Corresponding precipitation rate outputs from the same ensemble member are masked within 25° GCD of a storm center tracked in step 1. Here, all precipitation associated with the cyclone is retained while all precipitation not within 25° of a storm is set to zero.

To extract spatial information associated with ETCs we first filter a spatiotemporally continuous gridded dataset using NodeFileFilter:

```
NodeFileFilter --in_nodefile etc-all-traj.txt --in_fmt "lon,lat,slp" \
    --in_data_list B20TRC5CNBDRD.001.PRECT_list.txt \
    --out_data_list B20TRC5CNBDRD.001.PRECT_FILT_list.txt \
--var "PRECT" --bydist 25.0 --maskvar "mask"
```

A binary variable named 'mask' (as specified by argument `maskvar`) is also included in the filtered files for reference and can be used for offline masking and visualization.

As a last step, storm-centered composites are generated using the command:

```
NodeFileCompose --in_nodefile etc-strong-traj.txt --in_fmt "lon,lat,slp" \
--in_data_list B20TRC5CNBDRD.001.PRECT_FILT_list.txt \
    --out_data "composite_PRECT.nc" \
    --var "PRECT" --max_time_delta "2h" --op "mean" --dx 1.0 --resx 80
```

Here, only ETCs filtered above to have central SLP values below 990 hPa are composited. Although we can composite any 2D field, here we apply the compositing tool to precipitation filtered by NodeFileFilter. The argument `max_time_delta`
indicates that the data slice nearest in time to the tracked feature (within 2 hours) should be composited – this is useful when the discrete times from data and features are not exactly aligned. The arithmetic mean is calculated centered on the storm location (see section 2.4). The resulting stereographic composite has a grid spacing of 1° and a resolution of 80x80.

### 3.4.4 Results from compositing extratropical cyclone fields

Figure 5 shows the composited precipitation rate field (`PRECT`), along with analogously calculated composites of 850 hPa
temperature (`T850`) and integrated vapor transport (`IVT`). Total precipitation is largest near the storm center. Further, advection of warm, moist air wrapping cyclonically around the eastern side of the storm center is seen in the 850 hPa temperature field (composite wind vectors shown in black). Lastly, the collocation of high values of IVT and rising motion in the mid-troposphere (600 hPa omega contours shown in white) shows strong upward and poleward moisture advection associated with the warm conveyor belt, as previously shown in hand-compositing studies (e.g., Browning, 1986; Field and Wood, 2007).

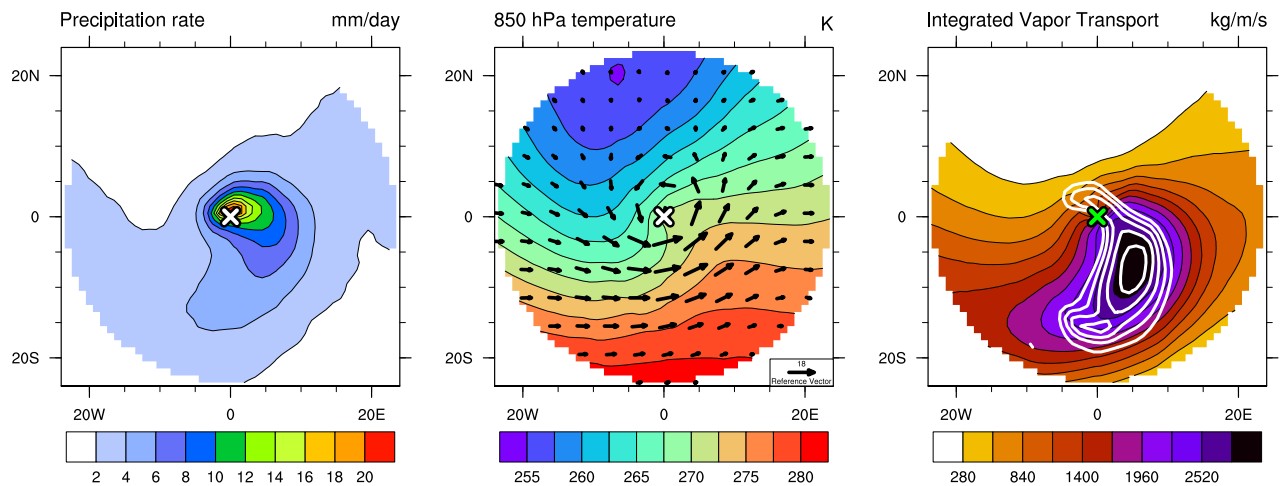

**Figure 5.** Composites of meteorological quantities centered on ETC storm center of all filtered storms with SLP less than or equal to 990 hPa in the first CESM-LENS historical member. Shown from left to right are precipitation rate (mm day$^{-1}$), 850 hPa temperature (K) with overlain 850 hPa wind vectors (m s$^{-1}$) and integrated vapor transport (g kg$^{-1}$) with overlain 600 hPa pressure velocity (omega) contours (every hPa hr$^{-1}$ starting at -2 hPa hr$^{-1}$) . Each composite includes 11,164 data points.

## 3.5 Atmospheric Rivers

Atmospheric rivers (ARs) are thin and long filamentary structures characterized by high integrated vapor transport (IVT; Payne et al., 2020). As found by Zhu and Newell (1998), ARs are responsible for approximately 90% of poleward vapor transport. Our goal in this section is to reproduce this result in 20 years of ERA5 reanalysis using the Tempest AR detection algorithm (Shields et al., 2018; Rhoades et al., 2020b, a; McClenny et al., 2020).

### 3.5.1 Step 1: Detect ridges in the IVT Field

As described in McClenny et al. (2020), the Tempest AR detection algorithm detects ARs as ridges in the IVT field, where IVT is defined pointwise as

$$\mathtt{IVT} = \sqrt{\mathtt{VIWVE}^2 + \mathtt{VIWVN}^2}. \tag{1}$$

Here we have adopted the nomenclature of ERA5 for vertically-integrated eastward vapor transport (VIWVE) and northward vapor transport (VIWVN). Ridge points are associated with high downward curvature, and identified as those points where the Laplacian of the IVT field is below a fixed threshold (here chosen to be $-2 \times 10^4$ kg m$^{-2}$s$^{-1}$rad$^{-2}$). These points are useful indicators of the presence of ARs because this threshold identifies either long and narrow features or localized maxima (which are subsequently filtered using a minimum area criterion). Here the Laplacian is calculated using 8 radial points at a distance of 10° GCD (as described in Appendix B); this large stencil on the Laplacian provides some smoothing of the field. Note that all field manipulation routines are handled by TE internally. The command line for this operation is as follows:

```
DetectBlobs --in_data_list ERA5_IVT_files.txt --out_list ERA5_AR_files.txt \
   --timefilter "6hr" \
   --thresholdcmd "_LAPLACIAN{8,10}(_VECMAG(VIWVE,VIWVN)),<=,-20000,0" \
   --minabslat 15 --geofiltercmd "area,>=,4e5km2"
```

The first three arguments in this command simply refer to the list of input files, output files and specify that data should be downsampled to 6-hourly timesteps. The meat of the operation is specified via the `thresholdcmd` argument, which uses the gridded data processor kernel built into TE to internally process the eastward and northward components of the integrated vapor transport (`VIWVE` and `VIWVN`, respectively) during the tagging operation. Specifically, the operation specified here identifies candidate grid points using a threshold on the Laplacian of the IVT. This command first calculates IVT using the vector magnitude operator, then calculates the Laplacian of the resulting field. Only points whose Laplacian is less than the threshold are retained. The last two arguments are then used to remove features too near the equator and those that are deemed too small: the latitude of each tagged grid point must be at least $15°$, and each blob must have a minimum area of $4 \times 10^5 \, \text{km}^2$. Such filtering criteria are typical for AR trackers (Shields et al., 2018).

### 3.5.2 Step 2: Filter out tropical cyclones

As noted in McClenny et al. (2020), tropical cyclones, which also tend to exhibit large values of IVT, are sometimes picked up as part of the detection procedure. Although their contribution to poleward IVT is small, it is nonetheless desirable to exclude TCs from this calculation. This can be done using the ERA5 TC tracks produced in section 3.2 to filter out points within a prescribed distance of each detected TC:

```
NodeFileFilter --in_nodefile ERA5_TC_tracks.txt --in_fmt "lon,lat,slp,wind,zs" \
   --in_data_list ERA5_AR_files.txt --out_data_list ERA5_AR_NFF_files.txt \
   --var "binary_tag" --bydist 8.0 --invert
```

Here the nodefile from ERA5 is specified by `in_nodefile` and `in_fmt`. The input list of files containing the AR binary masks, specified with `in_data_list` is the same as the output from DetectBlobs. The filtered output files are written to the filelist specified by `out_data_list`. The last two arguments here are key to the filtering procedure, specifying that the mask should include all points except those within 8 degrees GCD of each nodal feature.

Figure 6 shows the ERA5 integrated vapor transport (IVT) field on 2019-02-25 at 18:00 UTC, along with the outlines of AR objects detected using TE. On this date, an AR event on this date was responsible for flooding in California's Russian River basin (seen here intersecting the US West Coast). Dashed lines in this plot show the footprint of Super Typhoon Wutip at (139.75E, 15N) and Tropical Cyclone Pola at (175.5W, 14S), both of which have been excluded from the AR mask. Notably, Pola does not appear in IBTrACS until 2019-02-26 06:00:00 UTC.

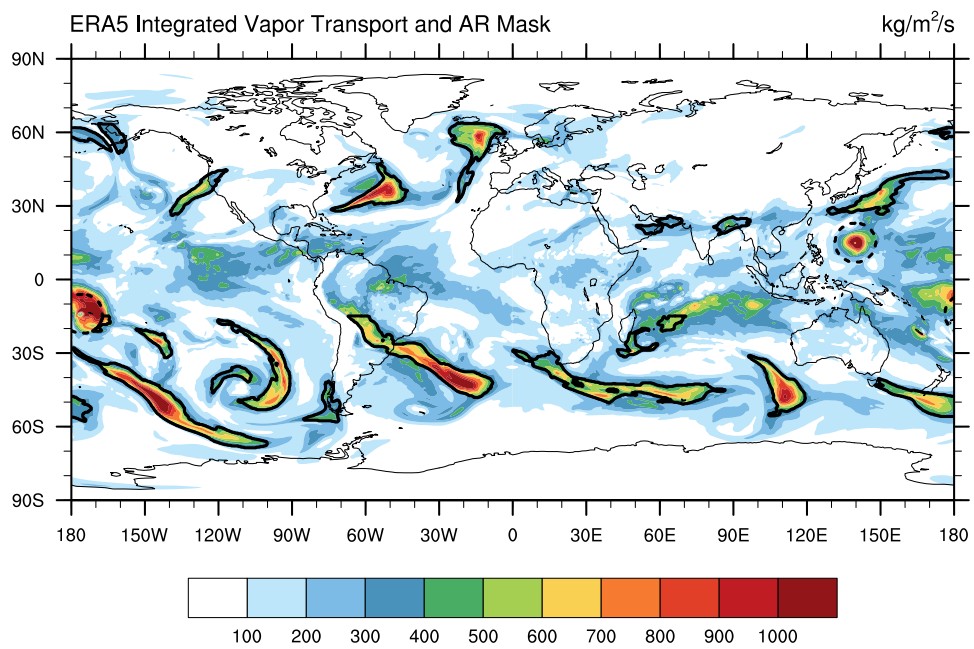

**Figure 6.** ERA5 integrated vapor transport (IVT) field with AR mask (black outlines) from 2019-02-25 18:00 UTC. Tropical cyclones that have been filtered from the AR mask are indicated with black dashed lines (8° radius GCD).

### 3.5.3 Step 3: Apply AR mask to northward vapor transport field

To now investigate AR and non-AR poleward moisture transport, we apply the mask generated in Step 2 to the `VIWVN` field (northward vapor transport). Here we leverage the VariableProcessor executable, which allows us to apply TE's built-in operations on a set of input files. Here the input file list `ERA5_VPIN.txt` is the same as `ERA5_AR_NFF_files.txt`, except with the corresponding ERA5 VIWVN file appended to each line. To perform the processing we apply the command:

```
VariableProcessor --in_data_list ERA5_VPIN.txt --out_data_list ERA5_VPOUT.txt \
  --timefilter "6hr" \
  --var "_PROD(binary_tag,VIWVN);_PROD(_DIFF(1,binary_tag),VIWVN)" \
  --varout "VIWVN_PW_AR,VIWVN_PW_NONAR"
```

The `var` argument here is specified to leverage TE's internal gridded variable processor. Since `binary_tag` only has value 0 or 1, the product of `VIWVN` and `binary_tag` will capture points within ARs, whereas the product of of `VIWVN` and `_DIFF(1,binary_tag)` will capture points not within ARs. These two variables are then written as `VIWVN_AR` and `VIWVN_NONAR` in the output file.

Once AR and non-AR northward IVT have been calculated on a gridpoint level, the final processing step is handled outside of TE. To do so we take the time average and zonal average of the fields produced by VariableProcessor using NCO's robust record-averaging (`ncra`) and weighted averager (`ncwa`) operators.

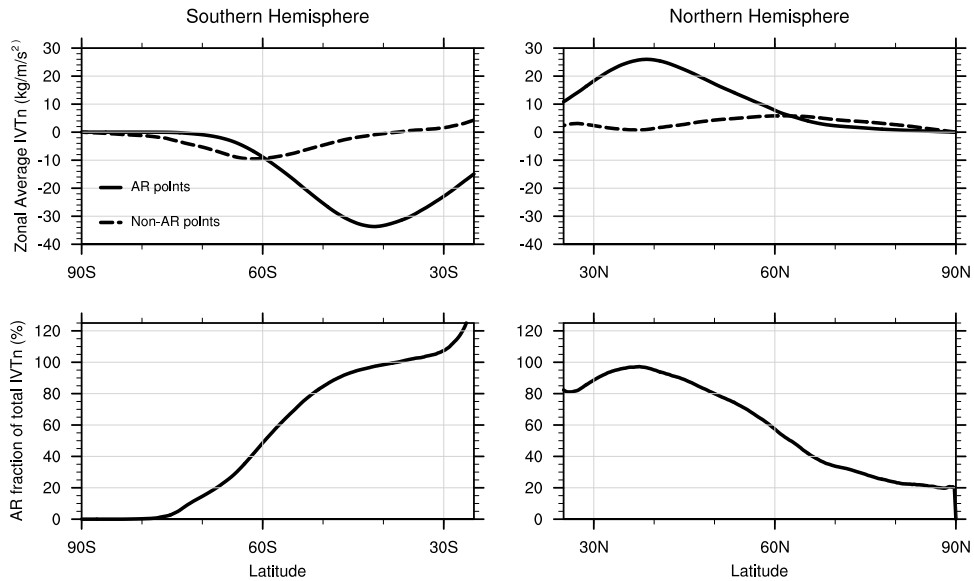

**Figure 7.** [top] Northward IVT (IVTn) from AR and non-AR grid points. [bottom] Fractional contribution to IVTn from AR points by latitude.

### 3.5.4 Results from calculation of northward vapor transport from AR and non-AR points

Figure 7 shows zonal mean northward IVT for AR and non-AR points (top row), along with the relative contribution to northward IVT from ARs (bottom row). Note that because it is a fractional quantity, the bottom row equivalently shows fractional contribution to poleward IVT. The top row here is complementary to Rutz et al. (2019) Figure 14 (middle), which was computed with 6-hourly MERRA2 data. The agreement between these two results is reassuring and confirms that the AR tracker employed in this section is consistent with other trackers. In the lower figure we see that the AR contribution to poleward transport around 45N and 45S is indeed close to the 90% value reported by Zhu and Newell (1998), although this contribution then decays precipitously at more poleward latitudes. Note, however, that this is in part because AR moisture transport is almost always poleward, whereas non-AR transport is a mix of both poleward and equatorward contributions.

The AR detector described in this section was run on the NERSC Cori supercomputer on two nodes with 32 threads per node (64 threads total). When run over the ERA5 monthly data from January 1979 through February 2020 at 6 hourly temporal resolution, with 494 monthly files, DetectBlobs required 34 minutes and 42 seconds. Again this run was largely I/O bound, with 66% of the total run time from file input. Approximately 13% of the total run time is spent applying the Laplacian operator, while 6% (2 minutes and 10 seconds) is spent constructing the Laplacian. Again using 64 threads, NodeFileFilter required 50 seconds while VariableProcessor required 14 minutes and 14 seconds.

### 3.6 Atmospheric Blocking

Our final example addresses the development of a climatology of atmospheric blocking frequency. Atmospheric blocking events are synoptic-scale weather phenomena characterized by persistent obstruction of the normal westerly flow and associated with heat waves, cold spells, flooding and drought (Glickman, 2012). In Pinheiro et al. (2019) (hereafter PUG19), several 2D algorithms were compared for the detection and characterization of blocking features; it was found that the identification algorithm of Dole and Gordon (1983) (hereafter DG83), which identifies blocks as anomalously high values of geopotential height at 500hPa (Z500), was a robust method for global block detection and characterization. In this section we will employ TE to generate a climatology of blocking events using the modified DG83 algorithm of PUG19 as applied to MERRA2 reanalysis data (Gelaro et al., 2017).

#### 3.6.1 Step 1: Generate the blocking threshold

Following PUG19, a grid point is defined as a candidate for being blocked if the Z500 field exceeds a threshold Z500 value. This threshold value must be specified as a function of latitude, longitude, and time, given the geographical and seasonal variations in Z500 climatology. PUG19 suggest a threshold value equal to the daily mean Z500 plus the maximum of 100 meters or 1.5 times the daily standard deviation of the Z500 field. Given that only 40 years of MERRA2 reanalysis are available, daily averaged data tends to be quite noisy and so Fourier smoothing is employed in time and space.

MERRA2 stores the 3D geopotential height variable in the `inst3_3d_asm_Np` dataset using variable name "H". For simplicity we assume that the input files contain a list of all files from this dataset from 1980/01/01-2020/06/30 (40.5 years). Within this dataset the 500hPa geopotential height variable can be specified by variable name `H(500hPa)`, where the vertical index is determined automatically by TE. The first step described in Pinheiro et al. (2019) is the construction of a Fourier-filtered long-term daily mean (LTDM) climatology of the Z500 field and its square. The Climatology executable is used in this step, and can be executed in parallel:

```
Climatology --in_data_list MERRA2_H_files.txt --out_data MERRA2_H_LTDM.nc  \
   --var "H(500hPa)" --period "daily" --type "mean" --missingdata
Climatology --in_data_list MERRA2_H_files.txt --out_data MERRA2_H2_LTDM.nc \
   --var "H(500hPa)" --period "daily" --type "meansq" --missingdata
```

Here the `missingdata` argument is needed since the 500hPa pressure surface sometimes falls below the ground in the vicinity of the Himalayas, which is indicated in MERRA2 with missing values. Note that, relevant to subsequent command lines, Climatology automatically prepends the descriptor "dailymean_" to the variable, so the final climatology is written to variable "dailymean_H".

We now calculate the standard deviation of the `H(500hPa)` field using the VariableProcessor:

```
VariableProcessor --in_data "MERRA2_H_LTDM.nc;MERRA2_H2_LTDM.nc" \
  --out_data "MERRA2_H_mean_stddev.nc" \
```

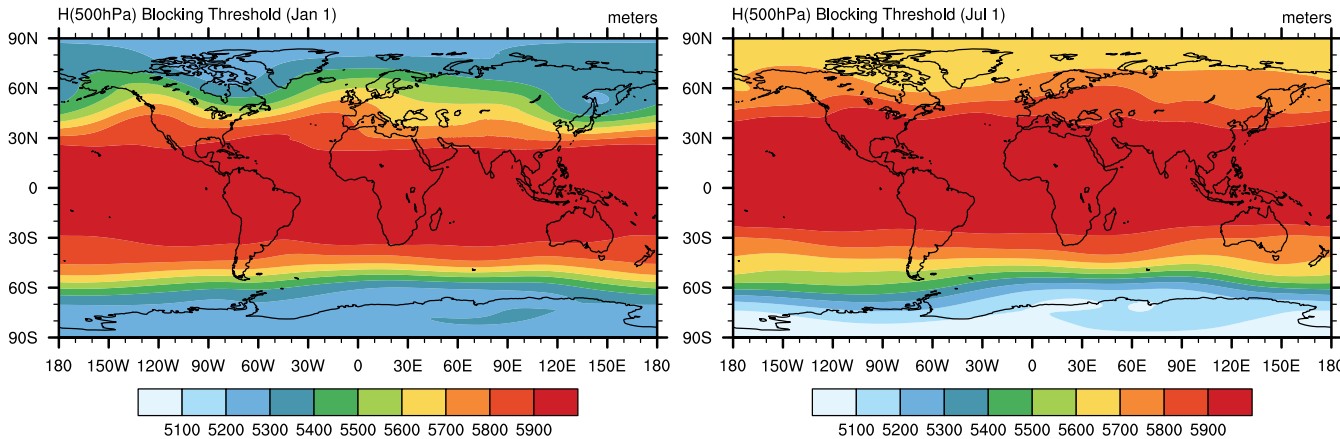

**Figure 8.** January 1st and July 1st MERRA2 blocking threshold generated with the command sequence in section 3.6.1.

```
    --var "dailymean_H,_SQRT(_DIFF(dailymeansq_H,_POW(dailymean_H,2)))" \
--varout "dailymean_H,stddev_H"
```

We then apply a 4-mode Fourier filter to both the `dailymean_H` and `stddev_H` fields across the `time` dimension, and a 2-mode Fourier filter to the `stddev_H` field in the zonal direction:

```
FourierFilter --in_data MERRA2_H_mean_stddev.nc \
    --out_data MERRA2_H_mean_stddev_timesmoothed.nc \
--var "dailymean_H,stddev_H" --dim "time" --modes 4
FourierFilter --in_data MERRA2_H_mean_stddev_timesmoothed.nc \
    --out_data MERRA2_H_mean_stddev_smoothed.nc \
    --var "stddev_H" --preserve "dailymean_H" --dim "lon" --modes 2
```

Finally the threshold is computed as $H500^* = \overline{H500} + \max\left(1.5 \times H500_{stddev}, 100\right)$ via the command line

```
VariableProcessor --in_data MERRA2_H_mean_stddev_smoothed.nc \
    --out_data MERRA2_threshold_H_filtered.nc \
    --var "_SUM(dailymean_H,_MAX(100.0,_PROD(1.5,stddev_H)))" --varout "threshold_H"
```

After performing these operations, a 365-day time series of the threshold field is obtained, plotted in Figure 8 on January 1st and July 1st. Note that this VariableProcessor operation could also have been performed using other climate data processing
software, such as NetCDF operators or a simple Python script; however, TE's support for parallelization over files allows for these computations to be performed rapidly on supercomputing systems.

### 3.6.2 Step 2: Identify regions of geopotential above the blocking threshold

With the LTDM blocking threshold in hand, we now define blocking features as sufficiently large contiguous regions where all points exceed the blocking threshold. As blocking is primarily a midlatitudinal feature, we also focus only on points between 25° N/S and 75° N/S.

The newly generated LTDM threshold file is appended to each line of the input data list so it can be accessed by DetectBlobs. The output file list has the same number of lines as this input list, but contains the output files that will contain the binary tags for tagged points.

```
DetectBlobs
  --in_data_list MERRA2_DB_files.txt --out_list MERRA2_blocktag_files.txt \
  --thresholdcmd "_DIFF(H(500hPa),threshold_H),>=,0,0" \
  --minabslat 25  --maxabslat 75 --geofiltercmd "area,>,1e6km2" --tagvar "block_tag"
```

This command tags points as candidates when 500 hPa geopotential height equals or exceeds the blocking threshold. We further remove candidate points equatorward of 25° N/S and poleward of 75° N/S and only retain contiguous regions whose area is at least $10^6 \mathrm{km}^2$.

### 3.6.3 Step 3: Enforce a minimum duration for blocks and build climatology

Besides being regions of anomalously high geopotential, blocking events must also be sufficiently persistent (Glickman, 2012). Although the AMS Glossary uses a typical duration of 7 days to indicate persistence, we follow PUG19 and retain events which last at least 5 days.

To determine the duration of individual events, we use StitchBlobs to connect blocking features in time, imposing that events are connected in time if they overlap by at least 20%. Features that do not persist for at least 5 days are discarded. The algorithm used here is described in detail in section 2.6. Here we use the following command:

```
StitchBlobs
  --in_list MERRA2_blocktag_files.txt --out_list MERRA2_blockid_files.txt \
  --var "block_tag" --mintime "5d" --min_overlap_prev 20 --flatten
```

The criteria here require that blobs exist for at least 5 days continuously, and that blobs are considered to be connected in time only when 20% of the area of the blocked object at the previous time step is covered by the new blob (specified by min_overlap_prev). The flatten argument indicates that only binary occurrence of a feature (0 or 1) should be recorded after stitching. If this argument was not specified, then each object would be assigned a unique global integer identifier, as described in section 2.6.

To build a seasonal climatology of blocking features, the blocking data from StitchNodes is post-processed. Since the presence of blocking is given with a binary indicator, this command is in effect calculating the fraction of timesteps where blocking occurs:

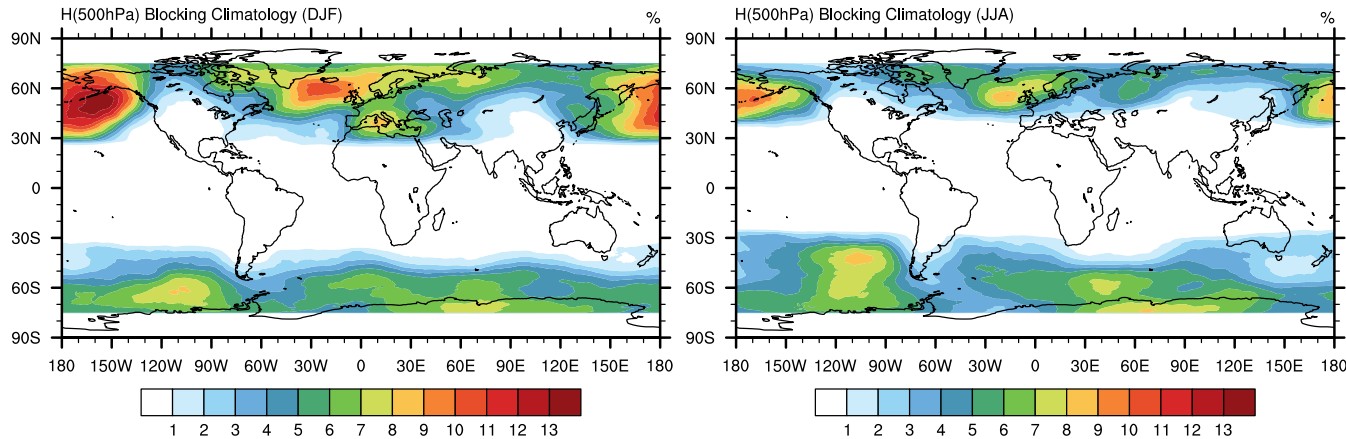

**Figure 9.** Percentage occurrence of blocking during (left) boreal winter and (right) boreal summer.

```
Climatology --in_data_list MERRA2_blockid_files.txt \
    --out_data MERRA2_blocking_climo.nc --var "object_id" --period "seasonal"
```

### 3.6.4 Results from the generated blocking climatology

The generated blocking climatology in the boreal winter (December-January-February) and boreal summer (June-July-August) is depicted in Figure 9. Results are generally in agreement with the climatology of Pinheiro et al. (2019), and show substantial wintertime blocking in the North Atlantic and Pacific. A snapshot of blocked regions on 2013-12-07 6Z is further depicted in Figure 10 (black outlines). The blocking feature present in the Northern Pacific at the time was associated with anomalous dry conditions in California and anomalous warmth in Alaska.

## 4 Conclusions

Automated feature tracking capabilities have been frequently and successfully employed throughout the literature to evaluate regional and global models, investigate the drivers and environments of extreme weather events, and understand future change in the statistics of atmospheric features. Feature trackers further provide an important mechanism for extracting relevant information from large climate datasets, including reanalysis and observational datasets, and climate model simulations. This is particularly important as the stakeholder needs for climate data associated with societally relevant impacts grow larger.

As there are significant overlaps across the core functionality of these trackers, there is a clear added benefit to integrating these kernels within a single framework. TempestExtremes (TE) is one such framework, with generalized kernels for identifying, characterizing, and analyzing both nodal and areal features. Although version 1 of TE was primarily focused on tropical and extratropical cyclones (Ullrich and Zarzycki, 2017), version 2 of TE has since added substantial new functionality for areal feature tracking, characterizing and compositing features, and more dataset-agnostic parameters and thresholds. TE focuses on

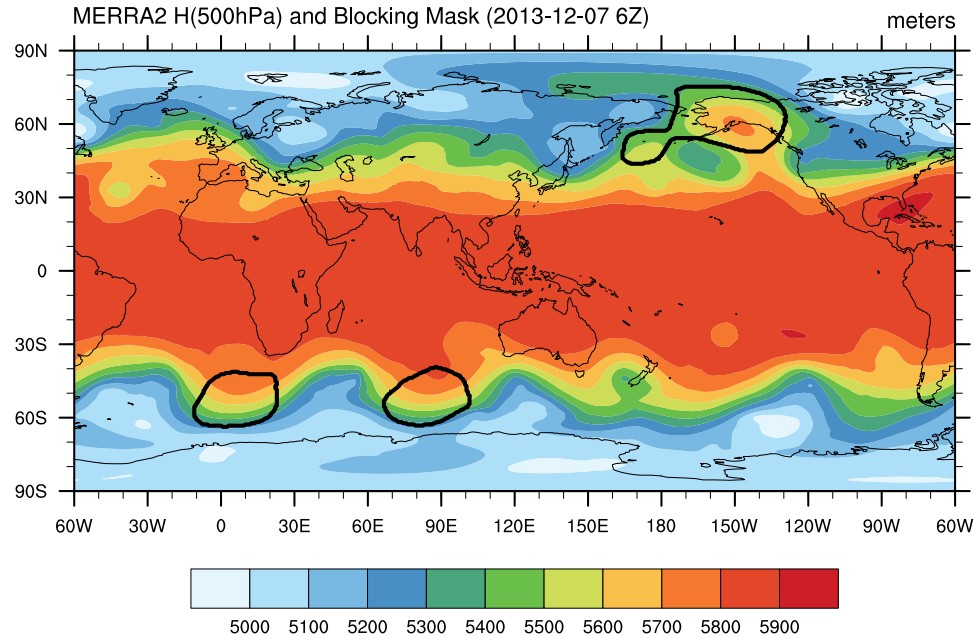

**Figure 10.** A snapshot of blocked regions in MERRA2 data detected on 2013-12-07 6Z (the three black outlines) atop the H(500hPa) field at this time.

high-throughput data processing, including continued MPI support for core executables. Such a framework has clear scientific relevance, enabling the development of feature catalogues, and addressing questions related to specific features, including those commonly associated with extreme weather. As TE further exposes all tuning parameters, users are able to easily investigate sensitivities of the tracker, or optimize the tracker for detecting particular features.

In this paper we have described some of the newer algorithmic kernels exposed by TE, and shown how these kernels can be composited to build robust tracking algorithms for many important atmospheric features. The tracking capability enables the probing of deeper scientific questions related to individual features. To demonstrate this functionality, TE was employed in section 3.3 for tracking of tropical cyclones (TCs) and calculated the fractional contribution of TC precipitation to total precipitation in each region. Ultimately, with the suite of trackers available from TE, a global climatology can be constructed that attributes total precipitation to features; i.e., using TE we showed that TCs contribute to 20-40% of total precipitation in the tropical regions of the Pacific and South Indian Ocean in both satellite observations and ERA5 reanalysis. In section 3.4, an analysis of the composited characters of extratropical cyclones in the CESM large ensemble was performed to understand climatological track density and meteorological fields, enabling evaluation of model performance and better communication of the relevant underlying processes. In section 3.5, a novel atmospheric river detection algorithm was developed using TE and validated against meridional moisture transport put forward in Zhu and Newell (1998). Finally, in section 3.6, TE was used to construct a seasonal climatology of atmospheric blocking. Notably, the data reductions demonstrated in these sections could support model evaluation via feature-specific and process-oriented metrics and diagnostics.

Nonetheless, TEv2.1 does have several limitations that may be addressed in future versions. At present, TE does not support detection of sub-grid-scale extrema (e.g., using harmonics as in Benestad and Chen (2006) or bicubics as in Murray and Simmonds (1991)), although this feature is largely necessitated by coarse spatial resolution inputs. TE also does not allow for extrapolation of the search position, as in some TC and ETC tracking schemes (Marchok, 2002). TE does not provide support for inline or offline percentile calculations, zonal/meridional averages, or time derivatives. Nor does v2.1 include support for

common calculus operators (e.g., relative vorticity, divergence, vector dot gradient, gradient magnitude), although experimental versions of these operators have been added in v2.2. It is also missing operations sometimes used for areal feature tracking, including dilation of areal features (Heikenfeld et al., 2019; Feng et al., 2018) and geometric operations sometimes used in AR tracking, including filtering of ARs with low width/length ratio (Mundhenk et al., 2016); support for these features is anticipated before v3.0. Additionally, as mentioned earlier in this paper, parallelism is presently only supported across files;

given that data products sometimes concatenate many times within a single file, support for parallelism within files is also desirable. In general, development of TE has been guided by the needs of its userbase, with many current features having been added by request; we anticipate this to continue into the future.

    It is expected that TE will continue to evolve to meet the needs of the scientific community. New kernels are already being investigated that encompass functionality present in other standalone trackers. A continued focus will be on maximizing TE's

robustness across datasets, so as to ensure the framework is useful for standalone users and operational modeling centers, or for comparative analysis across reanalysis products, multi-model ensembles (Eyring et al., 2016) and single model ensembles (Kay et al., 2015). Finally, new capabilities to perform direct evaluation of simulation products in TE are now being developed, using the characteristics of tracked features as evaluation metrics.

*Code availability.* The TempestExtremes v2.1 release is available from ZENODO at https://dx.doi.org/10.5281/zenodo.4385656. The GitHub
repository used for ongoing code development is available at https://github.com/ClimateGlobalChange/tempestextremes.

## Appendix A: Stereographic Projection

The stereographic projection is used in the construction of composites using NodeFileFilter. The equations used for projection are provided here for reference.

    The forward stereographic projection around a central point $(\lambda_0, \phi_0)$ is given by:

$$K = [1 + \sin\phi\sin\phi_0 + \cos\phi\cos\phi_0\cos(\lambda - \lambda_0)]^{-1} \tag{A1}$$

$$X(\lambda, \phi; \lambda_0, \phi_0) = K\cos\phi\sin(\lambda - \lambda_0) \tag{A2}$$

$$Y(\lambda, \phi; \lambda_0, \phi_0) = K[\cos\phi_0\sin\phi - \sin\phi_0\cos\phi\cos(\lambda - \lambda_0)] \tag{A3}$$

The inverse projection is given by:

$$\rho = \sqrt{X^2 + Y^2}, \tag{A4}$$

$$c = 2\arctan(\rho/2), \tag{A5}$$

$$\phi(X,Y;\lambda_0,\phi_0) = \begin{cases} \phi_0, & \text{if } \rho = 0, \\ \arcsin\left[\cos c \sin\phi_0 + (Y/\rho)\sin c \cos\phi_0\right], & \text{otherwise.} \end{cases} \tag{A6}$$

$$\lambda(X,Y;\lambda_0,\phi_0) = \begin{cases} \lambda_0, & \text{if } \rho = 0, \\ \lambda_0 + \arctan 2\left[X\sin c, \rho\cos\phi_0\cos c - Y\sin\phi_0\sin c\right], & \text{otherwise.} \end{cases} \tag{A7}$$

## Appendix B: Laplacian Operator

The stereographic discrete pointwise Laplacian operator defined in TE is constructed in a grid-independent manner using a discrete radial formulation. To begin, a set of $N$ initial sample points are generated using a ring of radius $R$ degrees around each grid point $\mathbf{X}_0$. Using a kd-tree-based implementation, the nearest grid points to each initial sample point are then selected to give a set of adjusted grid points $\mathbf{X}_n$ with $n = 1,\ldots,N$. For each of the initial sample points we then define the distance $D_n$ by the great-circle distance between grid point $X_0$ and $X_n$. The averaged Laplacian over a disc of radius $R/2$ is then computed discretely using the divergence theorem on the stereographic plane and a centered difference approximation for the radial derivative:

$$\frac{1}{\pi(R/2)^2}\int \nabla^2 q\,dA = \frac{4}{\pi R^2}\oint \nabla q \cdot dS \approx \frac{4}{\pi R^2}\sum_{n=0}^{N-1}\left(\frac{q_n - q_0}{D_n}\right)\left(\frac{\pi R}{N}\right) = \frac{4}{NR}\sum_{n=0}^{N-1}\frac{q_n - q_0}{D_n} \tag{B1}$$

where $q_n$ denotes the value of the field at $X_n$.

*Author contributions.* PAU wrote the software package with input from all involved authors and others in the community. CZ and KR advised on the algorithms for tracking tropical cyclones and extratropical cyclones. CZ further authored the extratropical cyclone section of the paper. EM advised on and tuned the algorithm for detecting and characterizing atmospheric rivers. MP developed the algorithm for detecting and characterizing atmospheric blocks. AS wrote the section on contributions of precipitation from tropical cyclones.

*Competing interests.* The authors declare that they have no competing interests.

*Acknowledgements.* The authors would like to acknowledge the feedback and bug reports from our many users, particularly Alan Rhoades, Karthik Belaguru, Vishnu Sasidharan Nair, Erica Bower, and Yumin Moon. This work has been supported by Department of Energy Office of Science award number DE-SC0016605, "A Framework for Improving Analysis and Modeling of Earth System and Intersectoral Dynamics at Regional Scales (HyperFACETS)," NASA award NNX16AG62G "TempestExtremes: Indicators of change in the characteristics

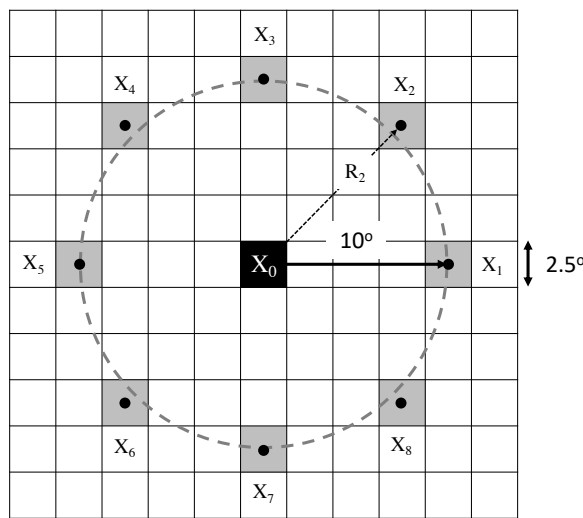

**Figure B1.** For a uniform (stereographic) grid with grid spacing of 2.5° GCD, an illustration of the grid points used in the calculation of the Laplacian with 8 radial points and radius 10° GCD. This operator is constructed in TE using notation `_LAPLACIAN{8,10}`. The central grid point is shaded black, and the modified centroids are shaded in gray.

of extreme weather," NASA award 80NSSC19K0717 "Quantifying the link between organized convection and extreme precipitation," and NOAA MAPP award NA19OAR4310288 "Future changes in the frequency of winter snowstorms and their impact on snowfall and snow water equivalent." This project is also supported by the National Institute of Food and Agriculture, U.S. Department of Agriculture, hatch
project under California Agricultural Experiment Station project accession no. 1016611. This research used resources of the National Energy Research Scientific Computing Center, a DOE Office of Science User Facility supported by the Office of Science of the U.S. Department of Energy under Contract No. DE-AC02-05CH11231.

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
