# Peer review of "TempestExtremes v2.1: A Community Framework for Feature Detection, Tracking and Analysis in Large Datasets"

_Geoscientific Model Development, 2020_

## Referee Comment (RC1)

**Review of gmd-2020-303 by Hristo G. Chipilski**

**Title:** TempestExtremes v2.1: A Community Framework for Feature Detection, Tracking and Analysis in Large Datasets

**Authors:** Paul A. Ullrich, Colin M. Zarzycki, Elizabeth E. McClenny, Marielle C. Pinheiro, Alyssa M. Stansfield, and Kevin A. Reed

**Suggested decision**: Accept with Minor Revisions

**General comments**

TempestExtremes (TE) is a framework for the identification and tracking of features in Earth system datasets. The underlying paradigm behind TE relies on the construction of abstract functions (kernels) that can be called directly from the command line and controlled via a highly configurable set of user parameters. In this work, the authors extend the original version of TE by carefully documenting all newly added kernels. Using several examples based on societally important meteorological features, they also demonstrate how one can configure TE for specific Earth system applications by sequentially combining relevant algorithm kernels. The robustness of the enhanced TE package is evident in its successful application to different geophysical features and the agreement of the obtained results with past studies. Because the upgraded version of TE generalizes previous tracking methods, the presented work constitutes an important contribution to the Earth system community as a whole. In view of this scientific merit and the high clarity of presentation, I strongly recommend the publication of the manuscript in GMD after the authors address my fairly minor comments below.

**Specific comments**

L31: If the authors wish to expand their list of areal feature tracking algorithms, they could give reference examples pertaining to convectively-generated outflow boundaries, such is in my 2018 model-based work (Chipilski et al. 2018) or precursor observation-based techniques, such as those of Uyeda and Zrnić (1986), Smith et al. (1989) and Delanoy and Troxel (1993). The complete references to these papers are as follows:

Chipilski, H. G., X. Wang, and D. B. Parsons, 2018: Object-based algorithm for the identification and tracking of convective outflow boundaries in numerical models. *Mon. Wea. Rev.*, **146**, 4179–4200, https://doi.org/10.1175/MWR-D-18-0116.1.

Delanoy, R. L., and S. W. Troxel, 1993: Machine Intelligent Gust Front Detection. *Lincoln Lab. J.*, **6**, 187–212.

Smith, S., A. Witt, M. D. Eilts, L. G. Hermes, D. Klingle-Wilson, S. Olson, and J. P. Stanford, 1989: Gust Front Detection Algorithm for the Terminal Doppler Weather Radar Part I: Current Status. *Proc. 3rd Intl. Conf. on the Aviation Weather System*, Anaheim, CA, 31.

Uyeda, H., and D. S. Zrnić, 1986: Automatic Detection of Gust Fronts. *J. Atmos. Ocean. Technol.*, **3**, 36–50, https://doi.org/10.1175/1520-0426(1986)003<0036:ADOGF>2.0.CO;2.

L44: Please define the abbreviation CF in CF-compliant.

L53: My advice is that you to not restrict to climate datasets only as TE can be applied with equal success to other types of Earth system datasets, e.g. outputs from Numerical Weather Prediction (NWP) models.

L62: "*except from DetectNodes and StichNodes*" – remove this as these kernels are described in Section 2.1.

L75: Here the authors mention that "*filtering of existing quantities*" is one of the capabilities present in NodeFileEditor. However, it is not immediately clear how the filtering in NodeFileEditor differs from the filtering operations in NodeFileFilter, so please add a brief clarification on this point.

L123-124 (discussion relevant to Figure 1): Could you please clarify in the text whether the ID of a merged object is equal to the smallest ID from the set of merging objects? In your example from Figure 1, objects 1 and 2 from time level 2 merge to a new object with an ID=1 at time level 3, i.e. $1=\min\{1,2\}$. Is this always the case?

L226-L227: Please elaborate on the meaning of "*This further provides an example of the ability of TE to evaluate functional relationships at run-time*" as it is not clear from the context what these functional relationships are and how you have defined them.

Figure 2: On L262-263, you state there is a high correlation between the algorithm-derived TC climatology and the observed TC tracks provided by IBTrACS. Is it possible to add the IBTrACS data as a subpanel in Figure 2 so that your readers confirm this conclusion? Ideally, I would like to see an algorithm-observation comparison similar to that shown in Figure 3.

L295: Briefly explain your choice of "*159 bins of width 0.125 degrees*" by either using a reference from the existing literature or a physically-based reasoning.

L393-L395: "*strong advection of warm, moist, equatorward air*" – avoid quantifying the strength of advection unless you decide to overlay the near-surface winds in Figure 5. Similarly, it is not possible to conclude "*that the heaviest ETC precipitation is associated with the warm conveyer belt*" in the absence of wind information.

L425: Here you could reword your subsection as "*Step 2: Create AR mask with NodeFileFilter*" in order to establish a better connection with the following "*Step 3: Apply AR mark to VIWVN*".

Figure 7: Replace "*northward*" with "*poleward*" to reflect that IVT refers to either the Northern or Southern Hemispheres.

**Technical corrections**

L143: Remove "*a*" in "*followed by a several examples*".

L144: "*subsequent employ*" should be changed to "*subsequent utilization*".

L146-L148: Remove the sentence starting with "*In each of these composite algorithms ...*" as you already mention this information earlier in your paragraph.

L209: Remove "*is*" in "*… our ERA5 data is comes …*".

L229: Avoid repeating "*output*"; e.g., you could replace the second "*output*" with "*written*".

L265: Please add the publication year to your Zarzycki et al. reference.

L545: It might be better to use "*integrating*" in lieu of "*developing*".

---

## Referee Comment (RC2)

**Review of "TempestExtremes v2.1: A Community Framework for Feature Detection, Tracking and Analysis in Large Datasets" in Geoscientific Model Development (gmd-2020-303)**

The authors present an updated version of the feature detection and tracking framework TempestExtremes (TE), after the first version of TE has been published in 2017. After reading that paper and the present manuscript, I am highly intrigued by TE for several reasons:

- Abstracting the grid by representing it as a graph allows for supporting unstructured grids and leveraging the capabilities and performance an optimized general-purpose graph library in a very elegant way.
- Given the prevalence of a relatively small number of operations/algorithms in many feature detection tools, a tool that implements them robustly and allows users to combine them definitely has merit, especially for relatively simple analyses.
- The interface as demonstrated in the examples seems to strike a good balance between consistency and relative simplicity on the one hand and flexibility to combine operators etc. on the other hand.

That being said, the present manuscript is not about TE per se, but specifically about version 2 and new features introduced since version 1. Crucially, TE now supports 2d (areal) features in addition to point (nodal) features, and is also capable of tracking them over time based on their spatial extent. This is an important step toward more general applicability of TE, especially high-resolution data that is becoming ever more prevalent today. New operators/kernels and tools to create feature profiles and composites are useful additions to TE as a comprehensive feature analysis framework.

As for the manuscript itself, it is on track but still a bit raw. After the initial submission was criticized by the editor for being too much like a software manual, the authors did improve the structure of the paper. However, in parts it still reads too much like a software tutorial (first major comment). Furthermore, while the authors demonstrate the wide applicability of TE, not enough is said about its limitations (second major comment), and while performance is frequently stressed as a point of focus, too little information about its actual performance is provided, especially compared to the tools with which it is compared scientifically (third major comment). These major comments are followed by a couple minor comments/suggestions (some of which are more crucial than others), which in turn are followed by a host of textual corrections/suggestions.

Once the major and (relevant) minor comments have been addressed, I am certain the manuscript will be in a good state for publication. I am looking forward to the revised version.

25 May 2021

**Major comments**

**1. Section 3**

The sections with example commands should be restructured such that the primary focus is on the physical phenomena, analyses and criteria, with the example commands and remarks on argument syntax coming second.

I do agree with the authors' response to the Editor in that it makes sense to include the commands with options in the paper because they illustrate the capabilities and limitations of TE in real applications. Given that the paper is about a specific version of TE, there is no danger of the commands becoming outdated (provided proper versioning by the developers), it may just be succeeded by later versions with a different interface.

Even though each example section is framed by a short introduction subsection in the beginning and a short discussion subsection in the end, overall, the section still reads too much like a tutorial than like a scientific paper. I propose the following changes:

First, while it makes sense subdividing the sections into individual subsection for each tool (e.g., "Step 1: DetectNodes"; "Step 2: StichNodes"), please change the titles to reflect the process/analysis step rather than the tool (e.g., "Step 1: Identification" or "Identify TCs"; "Step 2: Tracking" or "Track TCs over time"). "Step 1: Generate trajectories" is already a good example of this.

Second, please restructure the individual (sub-) sections. Now, they generally start with the command and go through the options while commenting on fields, thresholds and physical processes as needed. Instead, start by describing the goal of the step from a physical or algorithmic perspective, state what fields or algorithmic steps this requires, what thresholds are used, etc., and refer to any figures. Once the reader has the full picture of the analysis (step), show the TE command used to achieve it, and provide any crucial remarks on options (but leave out any others that are better suited for a separate user guide or tutorial, as suggested by the Editor).

Furthermore, in some instances the comparisons with literature results should be extended a bit; these are listed as minor comments.

**2. Limitations**

While the authors convincingly show that TE is able to replace many existing feature tracking tools, they hardly address its limits, i.e., what tools/algorithms are too complex or sophisticated to be easily replaced with TE.

An example for is the comparison with the IMILAST algorithms for extratropical cyclone identification/tracking (lines 356-357). The authors state that "other ETC detection algorithms analogous to those [in IMILAST] can be configured using TE's command line options." It would be interesting to know, however, how many of those algorithms can easily be replicated with TE, and whether there are some that are too sophisticated or complex, and in that case, what TE is lacking to do so.

These points can be briefly addressed in place like in the IMILAST example, and there should be at least a short discussion of the limitations of TE in the Discussion.

**3. Performance**

Throughout the paper, TE's ability to run efficiently in parallel on supercomputers is mentioned several times. However, not much is provided to substantiate this. On the one hand, when parallelism is mentioned, details on what is parallelized are usually missing (see minor comments). On the other hand, no performance measurements are provided. It would be helpful if at least a few comparisons with other tools could be added to give the reader at least some idea about the performance of TE beyond the qualitative descriptions currently in the manuscript.

**Minor comments**

Line 3: The equivalence of nodes and grid points is not necessarily clear if one is not aware of the graph representation of the grid in TE; also refer to nodal features as point features, either with "[both] point (nodal) [and areal features]" or "nodal (point)"

Lines 6-8: Extend with specific examples of both kernels and analyzed features, ideally including a scientific finding

Line 20: I am not sure how to understand "valid" in quotes; I'd suggest to just remove quotes, or alternatively provide a brief explanation like "valid trackers (in the sense of ...)"

Line 23: Same as on line 3 (abstract) regarding "nodal"

Line 33: Please briefly elaborate on the relevance of the correspondence between feature tracking and MapReduce, i.e., that it allows you to leverage existing work on this algorithms, as you wrote in the 2017 paper ("A key advantage of employing this framework is that substantial work has been undertaken to understand optimal strategies for parallelization of MapReduce-type algorithms (e.g., Prabhat et al., 2012) in order to mitigate bottlenecks associated with I/O and load balancing.")

Line 44: Please define "CF"

Lines 63-64: What do you mean by "applicability to either unstructured or structured grids"? Version 1 already handled unstructured grids, and therefore also structured grids. What has changed in this regard?

Line 90: I have a bit a hard time understanding what you mean by "snapshot". Intuitively, I understand a snapshot as the state of a full field at a certain moment, which makes the input fields to TE "snapshots" of the model state at that time step. However, "storm extraction" suggests that you understand "snapshot" as a certain part of a field, e.g., within a storm mask. My intuition may be wrong, but either way, could you please define what "snapshot" means in the context of this paper?

Line 100: What is parallelized with MPI? Space (domain decomposition), or something else?

Line 105: Could you briefly elaborate on the support of splitting/merging? What are the limits, in particular, with respect to the number of involved features -- only one-to/from-two, also one-to/from-many (and how many) or even many-to/from-many?

Line 115-124: I'd suggest not to use the variable names (min_overlap_prev etc.), given you don't use any variable names in the sections on the other executables. Try to reformulate in plain text or use more conventional (math-like) symbols similar to "$u_{(kt,max)}$" in Table 1. As it is now, this paragraph looks too much like a software manual.

Line 115-124: Could you provide an example where it is useful to constrain the maximum overlap (with max_overlap_*)? If there is none (apart from special cases), you could also just drop the maximum thresholds from this paragraph and stick to the minimum thresholds (the former are documented in the user manual, after all, if someone needs them).

Line 130: What is parallelized with MPI?

Line 145: Please add the data set(s) used for atmospheric river tracking, for consistency with all other examples

Lines 147-148: You already mentioned above why you show them ("effective at conveying ..."); only do this once

Lines 148: I don't understand "tunings" in this context; please reformulate this

Line 178: Replace "resolution of [ocean fronts]" by "resolving"

Line 204 and other commands: Add backslashes (line continuation) to the end of multiline commands that contain explicit newlines to make them valid shell commands

Lines 218-219: Please briefly elaborate on closed contours being desirable. If I understand correctly, you mean desirable over neighboring grid points to identify extrema, but it is hard to understand without having read the 2017 paper

Line 241: Consider defining the various TE file types ("nodefile" etc.), i.e., format, content, produced by what, in a table

Line 357: Do you mean a "more complex algorithm" than the one employed here, or more complex than those in IMILAST that can be represented with TE? Please clarify

Line 379: Please extend the comparison with other ETC trackers to a few sentences

Line 398: Replace "claimed in" by "found by"

Line 400: Replace "verify this claim" by "reproduce this result" (or "... finding")

Lines 402-403: The description of "ERA5_IVT_files.txt" definitely belongs in a user manual rather than here (see main comment on Section 3)

Line 505: Parallelized over what? Also, could you please elaborate on what exactly you are comparing here; would it for instance be very hard to achieve similar parallelism in Python, or are you just assuming that a "Python script" only contains a bunch of sequential commands (in which case you should at least precede it by "simple")?

Line 519: It would be helpful to accompany formulations like "stitch together blobs from each timeslice" with more generally-understandable equivalents, e.g., "The last step in block detection is to track the blocking areas over time, i.e., to stitch ... timeslice", or at least replace "[tracking of] blobs [in space]" by "blocking areas (blobs)"

Line: 549: Replace "command line arguments" with a synonym for functionality (that these are exposed as command line arguments is not relevant in this context)

Line 552: Remove "on the command line"; the main point is that they are exposed, how does not matter in this context

Line 554: Mention that this paper presents TE version 2 and reference the 2017 paper on version 1 again

Lines 568-569: Move "Using ... reanalysis" after the next sentence; also, cite the section, as you do for the subsequent results

Table 1, "radial_profile": What do you mean by "python-format array"? If you mean the dimension ordering, that should rather be C-style (i,j,k) or Fortran-style (k,j,i)

Figure 4: Add label to color bar (variable, unit)

**Typos, grammar, punctuation**

Line 2: Remove hyphen between "Earth"and "system"

Line 14: Move "such as large-scale meteorological patterns" out of parentheses; consider preceding "such" by comma and using semicolons between examples ("model performance (...); ... patterns (...); ...")

Lines 15-16: Replace "by which we can analyze" with "for analyzing"

Line 16: Remove parentheses around "and anticipated in the next decade"

Line 19: Replace "permit" by "provide"

Line 21: Replace "to [the choice of tracker]" by "with respect to"; remove "herein"

Line 22: Remove hyphen between "scientifically" and "driven"

Line 27: Remove hyphen between "regionally" and "relevant"

Line 38: Consider replacing "many" with, e.g., "a set of"

Line 40: Add comma after "i.e."

Line 42: Replace "command line(s)" by "commands" (or "command(s)")

Line 46: Replace ": namely" by "as", or turn "its kernels ..." into a standalone sentence

Line 49: Consider replacing "total" by "full"

Line 51: Consider removing "namely"

Line 60: Replace parentheses around "organized by executable" by a comma before "organized"

Lines 61-62: Remove the part of the sentence after "version 1.0". It's confusing to state that you won't emphasize DetectNodes and StichNodes (which sounds like you won't describe them), only to start with a section on exactly those two

Line 67: Replace "downselected" (?) by, e.g., "selected" or "detected" or "identified"

Line 69: Start new sentence after "chain"

Line 71: Replace "subsetting -- e.g." by "subsetting. For example,"

Line 72: Add commas after "e.g."s

Line 79: Add comma before "except"

Line 84: Replace dash after "information" by comma

Line 98: Replace "day" by "d" in unit

Line 99: Remove "using"

Line 104: Replace "in [sequential]" by "at"

Line 106: Consider replacing (or complementing) "recombining" with "merging"

Line 107: Add "is" before "illustrated"

Line 109: It looks like "(time id, blob id)" would be regular parenthetical expression, with the sentence finishing after "form". Consider putting it in quotes

Line 114: Replace "sequential times" by "subsequent time steps"

Line 126: Consider removing the quotes around "core"

Line 137: Replace "(NCO, Zender (2008))" with "(NCO; Zender, 2008)"

Line 139: Replace dash after "features" with comma

Line 141: Replace "command lines" with "commands" (twice); replace "these commands" with "they"

Line 144: Reformulate "and subsequent employ" (I'm not entirely sure what you want to say here)

Line 147: Replace "command lines" with "commands"

Line 154: Replace "used" by synonym (e.g., "employed") to reduce duplication

Line 173: Replace "criteria" with "criterion"

Line 188: Replace "[threshold]. This choice was made to [address]" with "in order to" and merge the two sentences

Line 202: Replace "minima" with "minimum"; replace "command line" with "command" (or "shell command")

Line 215: Replace "stringed together" with "separated"

Line 218: Replace "criteria" with "criterion"

Line 219: Replace "makes the criteria more robust" by "makes it more robust" or, e.g., "increases robustness"

Line 220: Replace "criteria" with "criterion"

Line 222: Replace "criteria" with "criterion"

Line 225: Replace "criteria" with "criterion"

Line 238: Replace "minima" with "minimum"

Lines 254-255: Replace "timeslices" by "time slices" (twice)

Line 295: Replace "consider" with "consist"

Line 315: Replace parentheses around "2017" by a comma after "Zarzycki"

Line 328: Replace "their process-level evaluation" by, e.g., "evaluating them at a process-level"

Line 334: Add "the" after "both"

Line 337: Replace "storm trajectories" by "storms"

Line 342: Replace "criteria" with "criterion"

Line 350: Replace "criteria" with "criterion" (unless there are indeed multiple DetectNodes criteria to satisfy)

Line 356: Replace "(IMILAST, Neu et al. (2013))" by "(IMILAST; Neu et al., 2013)"

Line 378: Replace "Step 1" with "step 1"

Line 389: Add "a" before and "of" after both "grid spacing" and "resolution"

Line 392: Replace "maximized" by, e.g., "largest" ("maximized" sounds like maximizing precipitation in the center was a compositing criterion)

Line 393: Move "equatorward" before "advection"

Line 395: Remove "other" before "hand-compositing"

Line 409: Replace "criteria" with "criterion"

Lines 423-424: Remove parentheses around "imposing .. 15" and reformulate (is "minimum area per blob" one of the isolated features, or is it a second criterion imposed on high-IVT features?)

Line 427: Add comma before "we can filter"

Lines 432-433: Either remove comma after "blobfiles" or add one also after "in_data_list"

Line 434: Add "at" before "18:00 UTC"

Line 444: Remove "lis"

Line 460: Replace "exhibits results" with, e.g., "produces results that"

Line 465: Add, e.g., "which are" between "blocking events," and "synoptic-scale" to prevent this from erroneously being read as a list at first

Lines 465-466: Replace "phenomenon" by "phenomena" ("blocking events" are plural); alternatively, reformulate to "[atmospheric blocking, which is a synoptic-scale weather] phenomenon ..."

Line 471: Reformulate "Z500 (geopotential height at 500 hPa)" to "compute the geopotential height anomaly at 500 hPa by applying the Z500 algorithm" (as it is, it looks like an inverted acronym definition)

Line 482: Replace "[below the] surface" by "ground"

Line 483: Simplify "and consequently ... employed" to, e.g., "which can cause problems", or just remove it

Line 505: Remove "using" before "a Python script"

Line 510: Replace "points to" by "contains"

Lines 515-516: Move "as candidate blob points" before "where 500 hPa"

Line 517: Replace "are" by "area"

Line 520: Replace "command line" by "command"

Line 527: Add comma before "then"

Line 549: Simplify "has further continued to remain focused" on, e.g., "focuses"

Lines 551-552: Replace parentheses around "including .. weather" by a comma after "specific features"

Line 552: Remove "features" after "those"

Line 560: Remove "in hand"

Line 564: Replace "results related to" with "against"

Line 566: Reformulate without "would" ("allow one to"? "enable"? ...)

Line 569: Reformulate "It will further continue to maximize its", e.g., "Continued focus will be on maximizing" or "A focus of TE will continue to be on maximizing" or so

Algorithm 2, caption: Replace "[closed contour] criteria" with "criterion"

Figure 1, caption: Add comma after "e.g."

Figure 2, caption: Reformulate sentence around "and applying" as there's something wrong

Figure 5, caption: Add "Shown" before "from left to right"; reformulate last sentence so it doesn't start with "11,164" (e.g., "Each composite includes ...")

Table 1, "eval_ace": Replace parentheses around "2000" by a comma after "al."

Table 1, "eval_acepsl": Add comma after "currently"

Table 1, "eval_ike": Replace "that [instantaneous]" by "the"; add "where" before "u_i"

Table 1, "radial_profile": Consider adding "by radial distance" (or similar) after "binning"

Table 1, "lastwhere": Replace "such as" by "e.g.,"; add comma before "identify"; add comma after "e.g."

Table 1, "lastwhere": What is returned; the array index? Or the distance?

Table 1, "value": Add comma before "extract"

Table 1, "max_closed_contour": Add comma before "determine"; replace "could be used to satisfy" by "satisfies"

Table 1, "region_name": Remove "containing"; replace dash after "latitude" by period and start a new sentence; add comma before "then the point"

---

## Author Comment (AC1)

**Review of gmd-2020-303 by Hristo G. Chipilski**

**Title:** TempestExtremes v2.1: A Community Framework for Feature Detection, Tracking and Analysis in Large Datasets
**Authors:** Paul A. Ullrich, Colin M. Zarzycki, Elizabeth E. McClenny, Marielle C. Pinheiro, Alyssa M. Stansfield, and Kevin A. Reed

**Suggested decision**: Accept with Minor Revisions

**General comments**

TempestExtremes (TE) is a framework for the identification and tracking of features in Earth system datasets. The underlying paradigm behind TE relies on the construction of abstract functions (kernels) that can be called directly from the command line and controlled via a highly configurable set of user parameters. In this work, the authors extend the original version of TE by carefully documenting all newly added kernels. Using several examples based on societally important meteorological features, they also demonstrate how one can configure TE for specific Earth system applications by sequentially combining relevant algorithm kernels. The robustness of the enhanced TE package is evident in its successful application to different geophysical features and the agreement of the obtained results with past studies. Because the upgraded version of TE generalizes previous tracking methods, the presented work constitutes an important contribution to the Earth system community as a whole. In view of this scientific merit and the high clarity of presentation, I strongly recommend the publication of the manuscript in GMD after the authors address my fairly minor comments below.

Thank you very much for your positive feedback and suggestions. We have extensively revised the manuscript in response to the two sets of reviewer comments and believe the resulting manuscript is much approved. Our response to individual comments can be found below.

**Specific comments**

L31: If the authors wish to expand their list of areal feature tracking algorithms, they could give reference examples pertaining to convectively-generated outflow boundaries, such is in my 2018 model-based work (Chipilski et al. 2018) or precursor observation-based techniques, such as those of Uyeda and Zrnić (1986), Smith et al. (1989) and Delanoy and Troxel (1993).
The complete references to these papers are as follows:

Chipilski, H. G., X. Wang, and D. B. Parsons, 2018: Object-based algorithm for the identification and tracking of convective outflow boundaries in numerical models. *Mon. Wea. Rev.*, **146**, 4179–4200, https://doi.org/10.1175/MWR-D-18-0116.1.
Delanoy, R. L., and S. W. Troxel, 1993: Machine Intelligent Gust Front Detection. *Lincoln Lab. J.*, **6**, 187–212.
Smith, S., A. Witt, M. D. Eilts, L. G. Hermes, D. Klingle-Wilson, S. Olson, and J. P. Stanford, 1989: Gust Front Detection Algorithm for the Terminal Doppler Weather Radar Part I: Current Status. *Proc. 3rd Intl. Conf. on the Aviation Weather System*, Anaheim, CA, 31.
Uyeda, H., and D. S. Zrnić, 1986: Automatic Detection of Gust Fronts. *J. Atmos. Ocean.*

*Technol.*, **3**, 36–50, https://doi.org/10.1175/1520-0426(1986)003<0036:ADOGF>2.0.CO;2.

Thank you for pointing out these additional references. We are confident there are many types of feature detection schemes we've missed in our brief review, but are happy to add mention of convective outflow boundaries and gust fronts. Thus we've included references to Chipilski et al. (2018) and Delanoy and Troxel (1993) in the paper.

L44: Please define the abbreviation CF in CF-compliant.

Fixed.

L53: My advice is that you to not restrict to climate datasets only as TE can be applied with equal success to other types of Earth system datasets, e.g. outputs from Numerical Weather Prediction (NWP) models.

Rephrased to:

*To the best of the authors' knowledge, no other comprehensive toolkit exists for general nodal and areal feature tracking in Earth system datasets.*

L62: "*except from DetectNodes and StichNodes*" – remove this as these kernels are described in Section 2.1.

Fixed.

L75: Here the authors mention that "*filtering of existing quantities*" is one of the capabilities present in NodeFileEditor. However, it is not immediately clear how the filtering in NodeFileEditor differs from the filtering operations in NodeFileFilter, so please add a brief clarification on this point.

The output of NodeFileEditor is a nodefile (text file containing trajectories). The output of NodeFileFilter is a NetCDF file containing filtered quantities. To be more specific, the opening of the "NodeFileEditor" section has been rephrased as:

*NodeFileEditor is a new addition to TE for editing nodefiles (i.e., output from StitchNodes). It includes options for (1) appending new details to trajectories, such as radial wind profiles or accumulated cyclone energy, (2) removing certain columns from nodefiles, or (3) filtering trajectories or points along a trajectory, e.g., when outside of a specific time interval.*

The opening of the "NodeFileFilter" section has been rephrased as:

*NodeFileFilter encapsulates algorithms for masking spatial data using nodefile information, i.e., effectively converting nodefiles into binary raster masks at each time slice and (optionally) applying them to available data.*

L123-124 (discussion relevant to Figure 1): Could you please clarify in the text whether the ID

of a merged object is equal to the smallest ID from the set of merging objects? In your example from Figure 1, objects 1 and 2 from time level 2 merge to a new object with an ID=1 at time level 3, i.e. 1=min{1,2}. Is this always the case?

*Merged objects are relabeled so that the global ids are sequential and without gaps. So the resulting global ID could be completely unrelated to the original tag. To clarify, the following sentence has been added to the text:*

*Note that global ids start at 1 and are consecutive thereafter; they are assigned only after connected components of the graph are identified, and as such are unrelated to the blob id on each time slice.*

L226-L227: Please elaborate on the meaning of "*This further provides an example of the ability of TE to evaluate functional relationships at run-time*" as it is not clear from the context what these functional relationships are and how you have defined them.

*This sentence has been rephrased to:*

*It is also an example of TE's ability to evaluate functions of meteorological fields at run-time.*

Figure 2: On L262-263, you state there is a high correlation between the algorithm-derived TC climatology and the observed TC tracks provided by IBTrACS. Is it possible to add the IBTrACS data as a subpanel in Figure 2 so that your readers confirm this conclusion? Ideally, I would like to see an algorithm-observation comparison similar to that shown in Figure 3.

*Added an additional panel to Fig. 2 that shows IBTrACS pointwise trajectories (in addition to top panel, which shows the TempestExtremes tracked cyclones in ERA5).*

L295: Briefly explain your choice of "*159 bins of width 0.125 degrees*" by either using a reference from the existing literature or a physically-based reasoning.

*Thank you for pointing this out. The following explanation has been added to the text:*

*The number of bins and bin width were chosen based on the horizontal grid spacing of the ERA5 wind data, which is approximately 31 km. The bin width of 0.125° was chosen to adequately sample points at this grid spacing to create the radial wind profiles. The number of bins was chosen to ensure the radial averaging extended out far enough from the TC center points to capture the storms' complete wind circulations.*

L393-L395: "*strong advection of warm, moist, equatorward air*" – avoid quantifying the strength of advection unless you decide to overlay the near-surface winds in Figure 5. Similarly, it is not possible to conclude "*that the heaviest ETC precipitation is associated with the warm conveyer belt*" in the absence of wind information.

*To address this, we have done two things. One, we have improved the composite panels, which now include wind barbs (to highlight advection) and vertical velocity contours (to highlight the*

WCB). We also plot IVT instead of the static low-level moisture field as this is more representative in showing poleward/upward advection of moisture on the eastern side (in the Northern Hemisphere) of the composite ETC.

In addition to the updated figure, the text as been modified to read:

*"Figure 5 shows the composited precipitation rate field (PRECT), along with analogously calculated composites of 850 hPatemperature (T850) and integrated vapor transport (IVT). Total precipitation is largest near the storm center. Further, advection of warm, moist air wrapping cyclonically around the eastern side of the storm center is seen in the 850 hPa temperature field (composite wind vectors shown in black). Lastly, the collocation of high values of IVT and rising motion in the mid-troposphere(600 hPa omega contours shown in white) shows strong upward and poleward moisture advection associated with the warm conveyor belt, as previously shown in hand-compositing studies (e.g., Browning, 1986; Field and Wood, 2007)."*

L425: Here you could reword your subsection as "*Step 2: Create AR mask with NodeFileFilter*" in order to establish a better connection with the following "*Step 3: Apply AR mark to VIWVN*".

Agreed. Something like this was also suggested by Reviewer 2 to make the paper read less like a technical document. As such all sections have been relabeled to emphasize the purpose rather than the operation.

Figure 7: Replace "*northward*" with "*poleward*" to reflect that IVT refers to either the Northern or Southern Hemispheres.

We believe in the top figures it is correct to use "northward", as values in the southern hemisphere are negative (i.e., poleward). In the bottom figures the term "northward" or "poleward" are interchangeable. The text is unchanged.

**Technical corrections**

L143: Remove "*a*" in "*followed by a several examples*".

Fixed.

L144: "*subsequent employ*" should be changed to "*subsequent utilization*".

Fixed.

L146-L148: Remove the sentence starting with "*In each of these composite algorithms …*" as you already mention this information earlier in your paragraph.

Fixed.

L209: Remove "*is*" in "*… our ERA5 data is comes …*".

Fixed.

L229: Avoid repeating "*output*"; e.g., you could replace the second "*output*" with "*written*".

Fixed.

L265: Please add the publication year to your Zarzycki et al. reference.

Fixed.

L545: It might be better to use "*integrating*" in lieu of "*developing*".

Fixed.

---

## Author Comment (AC2)

**Review of "TempestExtremes v2.1: A Community Framework for Feature Detection, Tracking and Analysis in Large Datasets" in Geoscientific Model Development (gmd-2020-303)**

The authors present an updated version of the feature detection and tracking framework TempestExtremes (TE), after the first version of TE has been published in 2017. After reading that paper and the present manuscript, I am highly intrigued by TE for several reasons:

- Abstracting the grid by representing it as a graph allows for supporting unstructured grids and leveraging the capabilities and performance an optimized general-purpose graph library in a very elegant way.
- Given the prevalence of a relatively small number of operations/algorithms in many feature detection tools, a tool that implements them robustly and allows users to combine them definitely has merit, especially for relatively simple analyses.
- The interface as demonstrated in the examples seems to strike a good balance between consistency and relative simplicity on the one hand and flexibility to combine operators etc. on the other hand.

That being said, the present manuscript is not about TE per se, but specifically about version 2 and new features introduced since version 1. Crucially, TE now supports 2d (areal) features in addition to point (nodal) features, and is also capable of tracking them over time based on their spatial extent. This is an important step toward more general applicability of TE, especially high-resolution data that is becoming ever more prevalent today. New operators/kernels and tools to create feature profiles and composites are useful additions to TE as a comprehensive feature analysis framework.

As for the manuscript itself, it is on track but still a bit raw. After the initial submission was criticized by the editor for being too much like a software manual, the authors did improve the structure of the paper. However, in parts it still reads too much like a software tutorial (first major comment). Furthermore, while the authors demonstrate the wide applicability of TE, not enough is said about its limitations (second major comment), and while performance is frequently stressed as a point of focus, too little information about its actual performance is provided, especially compared to the tools with which it is compared scientifically (third major comment). These major comments are followed by a couple minor comments/suggestions (some of which are more crucial than others), which in turn are followed by a host of textual corrections/suggestions.

Once the major and (relevant) minor comments have been addressed, I am certain the manuscript will be in a good state for publication. I am looking forward to the revised version.

We would like to thank the reviewer immensely for their very careful and constructive review. These comments have assisted immensely in improving the quality of this manuscript. Our response to individual comments can be found below.

**Major comments**

**1. Section 3**

The sections with example commands should be restructured such that the primary focus is on the physical phenomena, analyses and criteria, with the example commands and remarks on argument syntax coming second.

I do agree with the authors' response to the Editor in that it makes sense to include the commands with options in the paper because they illustrate the capabilities and limitations of TE in real applications. Given that the paper is about a specific version of TE, there is no danger of the commands becoming outdated (provided proper versioning by the developers), it may just be succeeded by later versions with a different interface.

Even though each example section is framed by a short introduction subsection in the beginning and a short discussion subsection in the end, overall, the section still reads too much like a tutorial than like a scientific paper. I propose the following changes:

First, while it makes sense subdividing the sections into individual subsection for each tool (e.g., "Step 1: DetectNodes"; "Step 2: StichNodes"), please change the titles to reflect the process/analysis step rather than the tool (e.g., "Step 1: Identification" or "Identify TCs"; "Step 2: Tracking" or "Track TCs over time"). "Step 1: Generate trajectories" is already a good example of this.

Second, please restructure the individual (sub-) sections. Now, they generally start with the command and go through the options while commenting on fields, thresholds and physical processes as needed. Instead, start by describing the goal of the step from a physical or algorithmic perspective, state what fields or algorithmic steps this requires, what thresholds are used, etc., and refer to any figures. Once the reader has the full picture of the analysis (step), show the TE command used to achieve it, and provide any crucial remarks on options (but leave out any others that are better suited for a separate user guide or tutorial, as suggested by the Editor).

Furthermore, in some instances the comparisons with literature results should be extended a bit; these are listed as minor comments.

Thank you for these excellent suggestions. We have followed this guidance in restructuring these sections.

**2. Limitations**

While the authors convincingly show that TE is able to replace many existing feature tracking tools, they hardly address its limits, i.e., what tools/algorithms are too complex or sophisticated to be easily replaced with TE.

An example for is the comparison with the IMILAST algorithms for extratropical cyclone identification/tracking (lines 356-357). The authors state that "other ETC detection algorithms analogous to those [in IMILAST] can be configured using TE's command line options." It would be interesting to know, however, how many of those algorithms can easily be replicated with TE, and whether there are some that are too sophisticated or complex, and in that case, what TE is lacking to do so.

These points can be briefly addressed in place like in the IMILAST example, and there should be at least a short discussion of the limitations of TE in the Discussion.

The following paragraph has been added to section 4, discussing TE's limitations:

*Nonetheless, TEv2.1 does have several limitations that may be addressed in future versions. At present, TE does not support detection of sub-grid-scale extrema (e.g., using harmonics as in Benestad and Chen (2006) or bicubics as in Murray and Simmonds (1991)), although this feature is largely necessitated by coarse spatial resolution inputs. TE also does not allow for extrapolation of the search position, as in some TC and ETC tracking schemes (Marchok, 2002). TE does not provide support for inline or offline percentile calculations, zonal/meridional averages, or time derivatives. Nor does v2.1 include support for common calculus operators (e.g., relative vorticity, divergence, vector dot gradient, gradient magnitude), although experimental versions of these operators have been added in v2.2. It is also missing operations sometimes used for areal feature tracking,including dilation of areal features (Heikenfeld et al., 2019; Feng et al., 2018) and geometric operations sometimes used in AR tracking, including filtering of ARs with low width/length ratio (Mundhenk et al., 2016); support for these features is anticipated before v3.0. Additionally, as mentioned earlier in this paper, parallelism is presently only supported across files given that data products sometimes concatenate many times within a single file, support for parallelism within files is also desirable. In general, development of TE has been guided by the needs of its userbase, and features are added as requested.*

**3. Performance**

Throughout the paper, TE's ability to run efficiently in parallel on supercomputers is mentioned several times. However, not much is provided to substantiate this. On the one hand, when parallelism is mentioned, details on what is parallelized are usually missing (see minor comments). On the other hand, no performance measurements are provided. It would be helpful if at least a few comparisons with other tools could be added to give the reader at least some idea about the performance of TE beyond the qualitative descriptions currently in the manuscript.

Thank you for pointing out this omission. One of the reasons why we don't focus on performance numbers is because these analyses tend to be heavily I/O bound. That is, most of the run-time of the code is taken up in file reads and writes rather than data processing. Thus there can be significant variability even within a single system attributed to how files are stored and indexed. However, we agree that some measures should be included in the paper and so have included text for TC tracking and AR detection.

The following text has been added in section 3.2.3:

*The TC detector described in this section was run on the NERSC Cori supercomputer on one node and using 32 threads. When run over the ERA5 data from January 1979 through February 2020 at 6 hourly temporal resolution, with 15,035 daily files, DetectNodes required 140 minutes run time. DetectNodes on Cori is strongly I/O bound, with reads from NetCDF files responsible for 81% of the total runtime. StitchNodes required 4 minutes and 55 seconds to process all 15,035 outputs from DetectNodes.*

And this text to section 3.5.4:

*The AR detector described in this section was run on the NERSC Cori supercomputer on two nodes with 32 threads per node (64 threads total). When run over the ERA5 monthly data from January 1979 through February 2020 at 6 hourly temporal resolution, with 494 monthly files, DetectBlobs required 34 minutes and 42 seconds. Again this run was largely I/O bound, with 66% of the total run time from file input. Approximately 13% of the total run time is spent applying the Laplacian*

*operator, while 6% (2 minutes and 10 seconds) is spent constructing the Laplacian. Again using 64 threads, NodeFileFilter required 50 seconds while VariableProcessor required 14 minutes and 14 seconds.*

**Minor comments**

Line 3: The equivalence of nodes and grid points is not necessarily clear if one is not aware of the graph representation of the grid in TE; also refer to nodal features as point features, either with "[both] point (nodal) [and areal features]" or "nodal (point)"

This has been changed to "nodal (i.e. pointwise)" on line 3 and line 23.

Lines 6-8: Extend with specific examples of both kernels and analyzed features, ideally including a scientific finding

Text has been added as follows:

*This paper describes the core algorithms (kernels) that have been added to the TE framework since version 1.0, including algorithms for editing pointwise trajectory files, composition of fields around nodal features, generation of areal masks via thresholding and nodal features, and tracking of areal features in time. Several examples are provided of how these kernels can be combined to produce composite algorithms for evaluating and understanding common atmospheric features and their underlying processes. These examples include analyzing the fraction of precipitation from tropical cyclones, compositing meteorological fields around extratropical cyclones, calculating fractional contribution to poleward vapor transport from atmospheric rivers, and building a climatology of atmospheric blocks.*

Line 20: I am not sure how to understand "valid" in quotes; I'd suggest to just remove quotes, or alternatively provide a brief explanation like "valid trackers (in the sense of ...)"

We agree the use of the word valid here is potentially misleading. We had implicitly inferred here that it is often difficult to define what exactly is a valid tracker for a particular feature, given that most atmospheric features are defined qualitatively. As a result we have removed the word "valid" from this sentence.

Line 23: Same as on line 3 (abstract) regarding "nodal"

This has been changed to "nodal (i.e. pointwise)" on line 3 and line 23.

Line 33: Please briefly elaborate on the relevance of the correspondence between feature tracking and MapReduce, i.e., that it allows you to leverage existing work on this algorithms, as you wrote in the 2017 paper ("A key advantage of employing this framework is that substantial work has been undertaken to understand optimal strategies for parallelization of MapReduce-type algorithms (e.g., Prabhat et al., 2012) in order to mitigate bottlenecks associated with I/O and load balancing.")

The following text has been added to the manuscript:

*By building a single framework for distributing time slices to different feature identification algorithms, then combining multiple features into a single dataset, we can avoid duplication of this*

*infrastructure across multiple trackers. Leveraging commonalities such as these enables improvements in algorithmic efficiency to be simultaneously administered to multiple trackers, and reduces redundancies from algorithmic validation and testing.*

Line 44: Please define "CF"

*Changed to:*

*Climate and Forecast compliant (CF-compliant)*

Lines 63-64: What do you mean by "applicability to either unstructured or structured grids"? Version 1 already handled unstructured grids, and therefore also structured grids. What has changed in this regard?

*Nothing has changed. We are simply noting here that support for unstructured grids has remained a design consideration. With this in mind, we have removed this text and added a bullet point to the list of TE's features on line 60:*

- *TE's algorithmic kernels are designed for arbitrarily grids, recognizing that climate models have largely moved away from latitude-longitude grids and towards quasi-uniform grids (Ullrich et al. 2017).*

Line 90: I have a bit a hard time understanding what you mean by "snapshot". Intuitively, I understand a snapshot as the state of a full field at a certain moment, which makes the input fields to TE "snapshots" of the model state at that time step. However, "storm extraction" suggests that you understand "snapshot" as a certain part of a field, e.g., within a storm mask. My intuition may be wrong, but either way, could you please define what "snapshot" means in the context of this paper?

*This paragraph has been changed as follows:*

*NodeFileCompose includes functionality for snapshotting fields around nodal features (i.e., at each time slice projecting fields onto the stereographic plane centered on a nodal feature) or compositing fields (i.e., averaging snapshots). In the same vein, it also includes functionality for snapshotting or compositing a particular geographic region when a feature is present. Stereographic composites are computed using Algorithm 4. The mathematical operators used for the local stereographic projection are given in Appendix A.*

Line 100: What is parallelized with MPI? Space (domain decomposition), or something else?

*The text now reads: DetectBlobs supports MPI-based parallelism over input files.*

Line 105: Could you briefly elaborate on the support of splitting/merging? What are the limits, in particular, with respect to the number of involved features -- only one-to/from-two, also one to/from-many (and how many) or even many-to/from-many?

*The algorithm is many-to/from-many. Any overlaps between time slices that satisfy the "overlap criteria" will result in an edge being generated in the connectivity graph. And features are then identified based on connected sub-graphs in that global connectivity graph. So potentially multiple mergers / splits could occur across one timestep. The following text has been added to this paragraph:*

*Since multiple edges could be generated to or from a feature on a given time slice, multiple*

*mergers or splits may occur simultaneously.  Finally, the components of the graph are each assigned a unique global id, with lower global ids corresponding to blobs that first appear at earlier times.  In Figure 1, feature 1 and 2 at time index 1, denoted (1,1) and (1,2), will both be assigned the same global id since they are connected at a later time.  Similarly, feature 1 and 2 at time index 3, denoted (3,1) and (3,2), are assigned the same global id since they were connected at an earlier time.*

Line 115-124: I'd suggest not to use the variable names (min_overlap_prev etc.), given you don't use any variable names in the sections on the other executables. Try to reformulate in plain text or use more conventional (math-like) symbols similar to "u_(kt,max)" in Table 1. As it is now, this paragraph looks too much like a software manual.

See following comment.

Line 115-124: Could you provide an example where it is useful to constrain the maximum overlap (with max_overlap_*)? If there is none (apart from special cases), you could also just drop the  maximum thresholds from this paragraph and stick to the minimum thresholds (the former are  documented in the user manual, after all, if someone needs them).

Agreed that constraining with maximum overlap is largely for special cases where we may want to break up blobs into multiple parts.  This paragraph has been rephrased as follows:

*By default, areal features are deemed to be connected in time if they share at least one grid point at sequential times (regardless of the area of that grid point).  For example, in Figure 1, areal regions (1,1) and (2,1) overlap in space and so are deemed to be connected.  If a stricter threshold on the overlap area is needed for blobs at sequential time slices to be deemed part of the same cluster, StitchBlobs provides arguments for minimum overlap between the current blob and blobs at the previous and/or next timestep.  In this example, blob tag (2,2) overlaps only 25% of the area of blob tag (1,2), meaning that (2,2) and (1,2) are deemed unconnected if the ``minimum previous overlap'' is greater than 25%.  On the other hand blob tag (1,2) covers 50% of the area of blob tag (2,2), so these two would be deemed unconnected if the ``minimum next overlap'' is greater than 50%.*

Line 130: What is parallelized with MPI?

Parallelization is over files.  This is now indicated in the text.

Line 145: Please add the data set(s) used for atmospheric river tracking, for consistency with all other examples

Added:  *atmospheric river tracking in ERA5*

Lines 147-148: You already mentioned above why you show them ("effective at conveying ..."); only  do this once

Removed.

Lines 148: I don't understand "tunings" in this context; please reformulate this

This sentence has been removed.  Tunings here referred to the specific values used for tracking.

Line 178: Replace "resolution of [ocean fronts]" by "resolving"

Fixed.

Line 204 and other commands: Add backslashes (line continuation) to the end of multiline commands that contain explicit newlines to make them valid shell commands
Thank you for the great suggestion -- the backslashes have been added.

Lines 218-219: Please briefly elaborate on closed contours being desirable. If I understand correctly, you mean desirable over neighboring grid points to identify extrema, but it is hard to understand without having read the 2017 paper.

This sentence has been changed to:

*As argued in Ullrich et al. 2017, closed contour criteria are a more physically grounded way of defining features since they can be employed for both discrete and continuous fields -- as opposed to, e.g., ``gridpoint maxima'' that are inherently sensitive to the dataset's grid structure and spacing.*

Line 241: Consider defining the various TE file types ("nodefile" etc.), i.e., format, content, produced by what, in a table

There is only one "TE file type" per se. These are now defined in section 2.1:

*DetectNodes and StitchNodes output trajectories in a format originally defined by the GFDL tropical cyclone tracker (TSTORMS; Vitart et al., 1997; Zhao et al., 2009). These files are generally referred to as nodefiles.*

There was previous mention to "blobfiles" as well, but those are essentially just NetCDF files containing a binary_tag variable. Thus the reference to "blobfile" has been replaced with "binary masks."

Line 357: Do you mean a "more complex algorithm" than the one employed here, or more complex than those in IMILAST that can be represented with TE? Please clarify

The following text has been added:

*Also, while a more complex algorithm could help eliminate cyclones that are tropical in nature (e.g., by using the --no_closed_contour argument to eliminate candidates with an upper level warm core), one is not applied here due to the relatively low resolution of CESM LENS.*

Line 379: Please extend the comparison with other ETC trackers to a few sentences

We have expanded the comparison with IMILAST trackers in two places, along with corresponding pointers to the manuscript for ease of comparison.

To highlight alternative approaches from an algorithmic standpoint:

*"These alternative approaches include tracking on low-level geopotential height (vorticity) minima (maxima), filtering based on spatial gradients, and removing candidate storms over higher terrain (see Table 1 in Neu et al. (2013))."*

To broadly compare North Atlantic storm tracks with IMILAST tracks:

*These results broadly match those of other ETC trackers depicted in Fig. 1 of Neu et*

*al. (2013), with a storm track belt extending across the North Atlantic centered on approximately 40-60°N latitude.*

Line 398: Replace "claimed in" by "found by"

Fixed.

Line 400: Replace "verify this claim" by "reproduce this result" (or "... finding")

Fixed.

Lines 402-403: The description of "ERA5_IVT_files.txt" definitely belongs in a user manual rather than here (see main comment on Section 3)

Fixed.

Line 505: Parallelized over what? Also, could you please elaborate on what exactly you are comparing here; would it for instance be very hard to achieve similar parallelism in Python, or are you just assuming that a "Python script" only contains a bunch of sequential commands (in which case you should at least precede it by "simple")?

Agreed with the inclusion of the word "simple" here.  While certainly achievable in Python, it would require substantial effort and technical expertise.  The text has been changed to:

*Note that this VariableProcessor operation could also have been performed using other climate data processing software, such as NetCDF operators or using a simple Python script; however, TE's support for parallelization over files allows for these computations to be performed rapidly on supercomputing systems.*

Line 519: It would be helpful to accompany formulations like "stitch together blobs from each timeslice" with more generally-understandable equivalents, e.g., "The last step in block detection is to track the blocking areas over time, i.e., to stitch ... timeslice", or at least replace "[tracking of] blobs [in space]" by "blocking areas (blobs)"

This sentence has been replaced with:

*The last step in block detection is to identify connected blobs, track those blobs in time, and exclude features that are not sufficiently persistent.*

Line: 549: Replace "command line arguments" with a synonym for functionality (that these are exposed as command line arguments is not relevant in this context)

Replaced with "parameters and thresholds"

Line 552: Remove "on the command line"; the main point is that they are exposed, how does not matter in this context

Replaced with:

*As TE further exposes all tuning parameters, users are able to easily investigate sensitivities of the tracker, or optimize the tracker for detecting particular features.*

Line 554: Mention that this paper presents TE version 2 and reference the 2017 paper on version 1 again

*Replaced with:*

*Although version 1 of TE was primarily focused on tropical and extratropical cyclones (Ullrich and Zarzycki, 2017), version 2 of TE has since added substantial new functionality for areal feature tracking, characterizing and compositing features, and more dataset-agnostic parameters and thresholds.*

Lines 568-569: Move "Using ... reanalysis" after the next sentence; also, cite the section, as you do for the subsequent results

Done.

Table 1, "radial_profile": What do you mean by "python-format array"? If you mean the dimension ordering, that should rather be C-style (i,j,k) or Fortran-style (k,j,i)

The array is only 1D as it is a function of radius. Here "python-formatted array" refers to it being a string that can be parsed by python into an array variable. This sentence has been replaced with:

*The output is expressed using python array syntax.*

Figure 4: Add label to color bar (variable, unit)

*This figure has been updated.*

**Typos, grammar, punctuation**

Line 2: Remove hyphen between "Earth"and "system"

Fixed.

Line 14: Move "such as large-scale meteorological patterns" out of parentheses; consider preceding "such" by comma and using semicolons between examples ("model performance (...); ... patterns (...); ...")

Fixed.

Lines 15-16: Replace "by which we can analyze" with "for analyzing"

Fixed.

Line 16: Remove parentheses around "and anticipated in the next decade"

Fixed.

Line 19: Replace "permit" by "provide"

Fixed.

Line 21: Replace "to [the choice of tracker]" by "with respect to"; remove "herein"

Fixed.

Line 22: Remove hyphen between "scientifically" and "driven"

Fixed.

Line 27: Remove hyphen between "regionally" and "relevant"

Fixed.

Line 38: Consider replacing "many" with, e.g., "a set of"

Fixed.

Line 40: Add comma after "i.e."

Fixed.

Line 42: Replace "command line(s)" by "commands" (or "command(s)")

Fixed.

Line 46: Replace ": namely" by "as", or turn "its kernels ..." into a standalone sentence

Fixed.

Line 49: Consider replacing "total" by "full"

Fixed.

Line 51: Consider removing "namely"

Fixed.

Line 60: Replace parentheses around "organized by executable" by a comma before "organized"

Fixed.

Lines 61-62: Remove the part of the sentence after "version 1.0". It's confusing to state that you won't emphasize DetectNodes and StichNodes (which sounds like you won't describe them), only to start with a section on exactly those two

Fixed.

Line 67: Replace "downselected" (?) by, e.g., "selected" or "detected" or "identified"

Fixed.

Line 69: Start new sentence after "chain"

Fixed.

Line 71: Replace "subsetting -- e.g." by "subsetting. For example,"

Fixed.

Line 72: Add commas after "e.g."s

Fixed.

Line 79: Add comma before "except"

Fixed.

Line 84: Replace dash after "information" by comma

Fixed.

Line 98: Replace "day" by "d" in unit

Fixed.

Line 99: Remove "using"

Fixed.

Line 104: Replace "in [sequential]" by "at"

Fixed.

Line 106: Consider replacing (or complementing) "recombining" with "merging"

This sentence has been rewritten as noted above.

Line 107: Add "is" before "illustrated"

Fixed.

Line 109: It looks like "(time id, blob id)" would be regular parenthetical expression, with the sentence finishing after "form". Consider putting it in quotes
Fixed.

Line 114: Replace "sequential times" by "subsequent time steps"

Fixed.

Line 126: Consider removing the quotes around "core"

Fixed.

Line 137: Replace "(NCO, Zender (2008))" with "(NCO; Zender, 2008)"

Fixed.

Line 139: Replace dash after "features" with comma

Fixed.

Line 141: Replace "command lines" with "commands" (twice); replace "these commands" with "they"

Fixed.

Line 144: Reformulate "and subsequent employ" (I'm not entirely sure what you want to say here)

This sentence has been replaced with:

*This is then followed by several examples from TE of feature-based tracking and subsequent analysis.*

Line 147: Replace "command lines" with "commands"

This sentence has since been removed, as noted above.

Line 154: Replace "used" by synonym (e.g., "employed") to reduce duplication

Fixed.

Line 173: Replace "criteria" with "criterion"

Fixed.

Line 188: Replace "[threshold]. This choice was made to [address]" with "in order to" and merge the two sentences

Fixed.

Line 202: Replace "minima" with "minimum"; replace "command line" with "command" (or "shell command")

Fixed.

Line 215: Replace "stringed together" with "separated"

Fixed.

Line 218: Replace "criteria" with "criterion"

This sentence has been rewritten.

Line 219: Replace "makes the criteria more robust" by "makes it more robust" or, e.g., "increases robustness"

This sentence has been rewritten.

Line 220: Replace "criteria" with "criterion"

Fixed.

Line 222: Replace "criteria" with "criterion"

Fixed.

Line 225: Replace "criteria" with "criterion"

Fixed.

Line 238: Replace "minima" with "minimum"

Fixed.

Lines 254-255: Replace "timeslices" by "time slices" (twice)

This term has been replaced throughout the manuscript.

Line 295: Replace "consider" with "consist"

Fixed.

Line 315: Replace parentheses around "2017" by a comma after "Zarzycki"

Fixed.

Line 328: Replace "their process-level evaluation" by, e.g., "evaluating them at a process-level"

Fixed.

Line 334: Add "the" after "both"

Fixed.

Line 337: Replace "storm trajectories" by "storms"

Fixed.

Line 342: Replace "criteria" with "criterion"

Fixed.

Line 350: Replace "criteria" with "criterion" (unless there are indeed multiple DetectNodes criteria to satisfy)

In this case there are multiple DetectNodes criteria to satisfy (closedcontourcmd and mergedist).
Line 356: Replace "(IMILAST, Neu et al. (2013))" by "(IMILAST; Neu et al., 2013)"

Fixed.

Line 378: Replace "Step 1" with "step 1"

Fixed.

Line 389: Add "a" before and "of" after both "grid spacing" and "resolution"

Fixed.

Line 392: Replace "maximized" by, e.g., "largest" ("maximized" sounds like maximizing precipitation in the center was a compositing criterion)

Fixed.

Line 393: Move "equatorward" before "advection"

Fixed.

Line 395: Remove "other" before "hand-compositing"

Fixed.

Line 409: Replace "criteria" with "criterion"

Fixed.

Lines 423-424: Remove parentheses around "imposing .. 15" and reformulate (is "minimum area per

blob" one of the isolated features, or is it a second criterion imposed on high-IVT features?)

Rewritten as:

*The last two arguments are then used to remove features too near the equator and those that are deemed too small: the latitude of each tagged grid point must be at least 15$^{o}$, and each blob must have a minimum area of 4 x 10$^{5}$ km$^{2}$.*

Line 427: Add comma before "we can filter"

Fixed.

Lines 432-433: Either remove comma after "blobfiles" or add one also after "in_data_list"

Rephrased to: *The input list of files containing the AR binary masks, ...*

Line 434: Add "at" before "18:00 UTC"

Fixed.

Line 444: Remove "lis"

Fixed.

Line 460: Replace "exhibits results" with, e.g., "produces results that"

Fixed.

Line 465: Add, e.g., "which are" between "blocking events," and "synoptic-scale" to prevent this from erroneously being read as a list at first

Fixed.

Lines 465-466: Replace "phenomenon" by "phenomena" ("blocking events" are plural); alternatively, reformulate to "[atmospheric blocking, which is a synoptic-scale weather] phenomenon ..."

Fixed.

Line 471: Reformulate "Z500 (geopotential height at 500 hPa)" to "compute the geopotential height anomaly at 500 hPa by applying the Z500 algorithm" (as it is, it looks like an inverted acronym definition)

Reformulated.

Line 482: Replace "[below the] surface" by "ground"

Fixed.

Line 483: Simplify "and consequently ... employed" to, e.g., "which can cause problems", or just remove it

Changed to:

*Here the missingdata argument is needed since the 500hPa pressure surface sometimes falls below the ground in the vicinity of the Himalayas, which is indicated in MERRA2 with missing values*

Line 505: Remove "using" before "a Python script"

Fixed.

Line 510: Replace "points to" by "contains"

Fixed.

Lines 515-516: Move "as candidate blob points" before "where 500 hPa"

This sentence has been rephrased as:

*This command tags points as candidates when 500 hPa geopotential height equals or exceeds the blocking threshold.*

Line 517: Replace "are" by "area"

Fixed.

Line 520: Replace "command line" by "command"

Fixed.

Line 527: Add comma before "then"

Fixed.

Line 549: Simplify "has further continued to remain focused" on, e.g., "focuses"

Fixed.

Lines 551-552: Replace parentheses around "including .. weather" by a comma after "specific features"

Fixed.

Line 552: Remove "features" after "those"

Fixed.

Line 560: Remove "in hand"

This sentence has been reformulated.

Line 564: Replace "results related to" with "against"

Fixed.

Line 566: Reformulate without "would" ("allow one to"? "enable"? ...)

This sentence has been rephrased as:

*Notably, the data reductions demonstrated in these sections could support model evaluation via feature-specific and process-oriented metrics and diagnostics.*

Line 569: Reformulate "It will further continue to maximize its", e.g., "Continued focus will be on maximizing" or "A focus of TE will continue to be on maximizing" or so

Reformulated as:

*A continued focus will be on maximizing TE's robustness across datasets, so as to ensure the framework is useful for standalone users and operational modeling centers, or for comparative analysis across reanalysis products, multi-model ensembles (Eyring et al., 2016) and single model ensembles (Kay et al., 2015).*

Algorithm 2, caption: Replace "[closed contour] criteria" with "criterion"

Fixed.

Figure 1, caption: Add comma after "e.g."

Fixed.

Figure 2, caption: Reformulate sentence around "and applying" as there's something wrong

Fixed.

Figure 5, caption: Add "Shown" before "from left to right"; reformulate last sentence so it doesn't start with "11,164" (e.g., "Each composite includes ...")

Fixed.

Table 1, "eval_ace": Replace parentheses around "2000" by a comma after "al."

Fixed.

Table 1, "eval_acepsl": Add comma after "currently"

Fixed.

Table 1, "eval_ike": Replace "that [instantaneous]" by "the"; add "where" before "u_i"

Fixed; reformulated.

Table 1, "radial_profile": Consider adding "by radial distance" (or similar) after "binning"

Fixed.

Table 1, "lastwhere": Replace "such as" by "e.g.,"; add comma before "identify"; add comma after "e.g."

Reworded.

Table 1, "lastwhere": What is returned; the array index? Or the distance?

If it is a normal array with integer indices the index is returned. The radial profile uses indices that are based on distance, so distance is returned when this command is combined with distance. The language has been clarified.

Table 1, "value": Add comma before "extract"

Fixed.

Table 1, "max_closed_contour": Add comma before "determine"; replace "could be used to satisfy"

by "satisfies"

Fixed.

Table 1, "region_name": Remove "containing"; replace dash after "latitude" by period and start a new sentence; add comma before "then the point"

Fixed.

---

## Referee Report (RR1)

Line 31: Remove comma after "depressions"

Line 39: "Assessed independent [of one another]" sounds odd to me; consider replacing with "independently" or "assessed as independent"

Line 47: Consider not repeating "TE" at the beginning of each bullet points by reformulating them in imperative form: "Encapsulate Kernels in ...", "Abstract many of the ...", Directly address the need ..." etc.

Lines 51-52: Move "compliant" after acronym definition: "Climate and Forecast (CF) compliant"

Line 54: Consider putting "where appropriate" in parentheses

Line 64: Consider reformulating "In section 3, we present" to "Section 3 present", equivalently to the previous sentence

Line 65: Remove "together" after "combined"

Line 73: Consider italicizing the executable names at the beginning of Sections 2.1-2.6 and throughout the paper, as you did in the list items in Section 2.7

Line 74: Remove "first" before "selected"

Line 75: Add "processing" or something similar before "chain"; remove "together" after "features"

Lines 83-85: Replace commas before "(2)" and "or (3)" by semicolons

Table 1: Capitalize "python" in description of "radial_profile"

Line 116: Add quotes around "2D space + 1D time" and add "3D" before "[single] object"

Lines 125-126: Replace "feature [1 and 2]" by "features" (2x)

Line 151: Consider removing "Selected" from title to avoid duplication with next sentence

Line 208: Replace "applied" with, e.g., "performed" or "carried out"

Line 222: Remove "are" before "points"

Line 223: Hyphenize "upper level"

Line 239: Hyphenize "closed contour [criteria]" (2x)

Line 242: Remove parentheses around "Pa"

Line 243: Remove hyphens from "great-circle-distance"

Line 250: Use acronym for "great circle distance"

Line 251: Move "argument" before "outputcmd"

Line 256: Remove "together"

Line 268: Add comma after "Nonetheless"

Line 274: Add "a" before "TC"

Line 286: Replace "ERA5 tracked storms" with "storms tracked in ERA5" (or "tracked storms in ERA5")

Line 302: Remove "r8" after "radius" as it is only defined in the following section

Line 305: Replace "outer size" by, e.g., "outer boundary", or simply by "size"

Line 306: Replace "[runs] from" with "with"

Line 307: Consider replacing "This [paper]" by "That" as "this" may refer to the paper one is currently reading

Figure 2: Replace "Category is" by "Categories are" (or "The categories are") in caption

Line 326: Replace "output" with "written"

Line 327: Add "the" before "bin width"

Lines 328-329: Consider simplifying "was chosen to adequately sample" to "adequately samples" and "was chosen to ensure" to "ensures"

Line 331: Add "the" before "storm" or remove "the" before "outer radius"

Line 332: Replace "in [grid points]" by "at"

Line 342: Add space between time and "UTC" (here and everywhere else)

Line 349: Remove quotes around "by distance"

Line 351: Replace "through the employ" by "by employing" or "through the application"

Line 353: Replace "percent [contribution]" by "relative" or "percentage"

Line 361: Acronym "ETC" has already been defined

Line 371: Consider removing "other" before "features" so as not to imply that topographic lows are non-meteorological phenomena

Lines 376-377: Consider reformulating "geopotential (vorticity) minima (maxima)", which I find a bit confusing; e.g., to "geopotential minima or vorticity maxima"

Line 379: Consider replacing "one is not [applied]" by "none is"

Line 381: Hyphenize "higher resolution"

Line 397: Replace "Further" with "Furthermore"

Line 402: Use previously defined acronym for "sea level pressure"

Line 418: Consistently (non-) capitalize "step 1" etc. throughout the paper/section (I'd suggest lowercase)

Line 437: Consider adding "grid points" after "80x80"

Line 460: Remove "line" after "command"

Line 465: Remove "in this command simply"

Line 466: Consider removing "meat of the" in order to simplify the sentence and make it less colloquial

Line 483: Add comma after "in_data_list" or remove that before "specified"

Line 484: Replace "filelist" with "file" (or at least add a space)

Line 487: Remove one of the two "on this date"

Line 494: Add "in" before "ERA5_VPIN.txt" and "ERA5_AR_NFF_files.txt"

Line 506: Remove hyphen in "record-averaging" and, if possible, use "averaging" (or "averager") for both "ncra" and "ncwa"

Line 526: Replace "features; it" with "features. It" (new sentence)

Line 554: Remove "lines" after "command"

Line 564: Remove "line" after "command"

Lines 596-598: These criteria have already been specified above the command

Line 606: Remove "generated"

Line 609: Specify date with "UTC" (not "Z") as in the rest of the paper

Line 618: Replace "across" with "in"

Line 620: Add, e.g., "(point)" after "nodal" (as in the beginning of the paper); replace "Although [version 1]" with "While"

Line 623: Remove "continued" and consider adding "to parallelize file system operations" or something similar to the end of the sentence (to highlight what it parallelized)

Line 630: Replace "calculated" by "calculating" or "computing"

Line 631: Replace "available from" with "provided by" or "based on" (whichever you mean)

Line 632: Replace "i.e." by "e.g."

Line 641: Replace "sub-grid-scale" with "subgrid-scale"

---

## Author Response (AR2)

**Review of "TempestExtremes v2.1: A Community Framework for Feature Detection, Tracking and Analysis in Large Datasets" in Geoscientific Model Development (gmd-2020-303)**

The authors have provided comprehensive replies to the reviewers and incorporated their suggestions into the manuscript, which has considerably improved the quality of the paper in terms of both content and presentation. I now recommend the paper for publication. Attached is a list of textual corrections/suggestions that the authors may take into account at their discretion during a final textual revision, and to which I am NOT asking for a reply.

We would like to thank the reviewer for a second round of very comprehensive suggestions. These reviews have made the lead author strongly question their own competence when it comes to scientific writing.

Line 31: Remove comma after "depressions"

Fixed.

Line 39: "Assessed independent [of one another]" sounds odd to me; consider replacing with "independently" or "assessed as independent"

Changed to "independently assessed".

Line 47: Consider not repeating "TE" at the beginning of each bullet points by reformulating them in imperative form: "Encapsulate Kernels in ...", "Abstract many of the ...", Directly address the need ..." etc.

Fixed.

Lines 51-52: Move "compliant" after acronym definition: "Climate and Forecast (CF) compliant"

Fixed.

Line 54: Consider putting "where appropriate" in parentheses

Fixed.

Line 64: Consider reformulating "In section 3, we present" to "Section 3 present", equivalently to the previous sentence

Fixed.

Line 65: Remove "together" after "combined"

Fixed.

Line 73: Consider italicizing the executable names at the beginning of Sections 2.1-2.6 and throughout the paper, as you did in the list items in Section 2.7

Fixed.

Line 74: Remove "first" before "selected"

Fixed.

Line 75: Add "processing" or something similar before "chain"; remove "together" after "features"

Fixed.

Lines 83-85: Replace commas before "(2)" and "or (3)" by semicolons

Fixed.

Table 1: Capitalize "python" in description of "radial_profile"

Fixed.

Line 116: Add quotes around "2D space + 1D time" and add "3D" before "[single] object"

Fixed.

Lines 125-126: Replace "feature [1 and 2]" by "features" (2x)

Fixed.

Line 151: Consider removing "Selected" from title to avoid duplication with next sentence

The first sentence has been changed to "several examples"

Line 208: Replace "applied" with, e.g., "performed" or "carried out"

Fixed.

Line 222: Remove "are" before "points"

Fixed.

Line 223: Hyphenize "upper level"

Fixed.

Line 239: Hyphenize "closed contour [criteria]" (2x)

Left unchanged. It's unclear why "closed contour" should be hyphenated here but not elsewhere in the text.

Line 242: Remove parentheses around "Pa"

Fixed.

Line 243: Remove hyphens from "great-circle-distance"

Fixed.

Line 250: Use acronym for "great circle distance"

Fixed.

Line 251: Move "argument" before "outputcmd"

Fixed.

Line 256: Remove "together"

Fixed.

Line 268: Add comma after "Nonetheless"

Fixed.

Line 274: Add "a" before "TC"

Fixed.

Line 286: Replace "ERA5 tracked storms" with "storms tracked in ERA5" (or "tracked storms in ERA5")

Fixed.

Line 302: Remove "r8" after "radius" as it is only defined in the following section

Fixed.

Line 305: Replace "outer size" by, e.g., "outer boundary", or simply by "size"

Fixed.

Line 306: Replace "[runs] from" with "with"

Fixed.

Line 307: Consider replacing "This [paper]" by "That" as "this" may refer to the paper one is currently reading

Changed to "They [further compared]"

Figure 2: Replace "Category is" by "Categories are" (or "The categories are") in caption

Fixed.

Line 326: Replace "output" with "written"

Fixed.

Line 327: Add "the" before "bin width"

Fixed.

Lines 328-329: Consider simplifying "was chosen to adequately sample" to "adequately samples" and "was chosen to ensure" to "ensures"

Fixed.

Line 331: Add "the" before "storm" or remove "the" before "outer radius"

Fixed.

Line 332: Replace "in [grid points]" by "at"

Fixed.

Line 342: Add space between time and "UTC" (here and everywhere else)

Fixed.

Line 349: Remove quotes around "by distance"

Fixed.

Line 351: Replace "through the employ" by "by employing" or "through the application"

Fixed.

Line 353: Replace "percent [contribution]" by "relative" or "percentage"

Fixed.

Line 361: Acronym "ETC" has already been defined

Fixed.

Line 371: Consider removing "other" before "features" so as not to imply that topographic lows are non-meteorological phenomena

Fixed.

Lines 376-377: Consider reformulating "geopotential (vorticity) minima (maxima)", which I find a bit confusing; e.g., to "geopotential minima or vorticity maxima"

Fixed.

Line 379: Consider replacing "one is not [applied]" by "none is"

Fixed.

Line 381: Hyphenize "higher resolution"

Fixed.

Line 397: Replace "Further" with "Furthermore"

Fixed.

Line 402: Use previously defined acronym for "sea level pressure"

Not changed.  We have previously used PSL, SLP and MSL to denote sea level pressure, as it is inconsistent among datasets.  So we think it is best here to avoid any acronym.

Line 418: Consistently (non-) capitalize "step 1" etc. throughout the paper/section (I'd suggest lowercase)

Fixed throughout (with lowercase).

Line 437: Consider adding "grid points" after "80x80"

Fixed.

Line 460: Remove "line" after "command"

Fixed.

Line 465: Remove "in this command simply"

Fixed.

Line 466: Consider removing "meat of the" in order to simplify the sentence and make it less colloquial

Changed to "[The] gridpoint-level filtering [operation is specified…]"

Line 483: Add comma after "in_data_list" or remove that before "specified"

Fixed.

Line 484: Replace "filelist" with "file" (or at least add a space)

Fixed.

Line 487: Remove one of the two "on this date"

Fixed.

Line 494: Add "in" before "ERA5_VPIN.txt" and "ERA5_AR_NFF_files.txt"

Changed "file list" to "file".

Line 506: Remove hyphen in "record-averaging" and, if possible, use "averaging" (or "averager") for both "ncra" and "ncwa"

Fixed.

Line 526: Replace "features; it" with "features. It" (new sentence)

Fixed.

Line 554: Remove "lines" after "command"

Fixed.

Line 564: Remove "line" after "command"

Fixed.

Lines 596-598: These criteria have already been specified above the command

Removed.

Line 606: Remove "generated"

Subsection title changed to "Blocking climatology results"

Line 609: Specify date with "UTC" (not "Z") as in the rest of the paper

Fixed.

Line 618: Replace "across" with "in"

Fixed.

Line 620: Add, e.g., "(point)" after "nodal" (as in the beginning of the paper); replace "Although [version 1]" with "While"

Fixed.

Line 623: Remove "continued" and consider adding "to parallelize file system operations" or something similar to the end of the sentence (to highlight what it parallelized)

Fixed.

Line 630: Replace "calculated" by "calculating" or "computing"

Fixed.

Line 631: Replace "available from" with "provided by" or "based on" (whichever you mean)

Fixed.

Line 632: Replace "i.e." by "e.g."

Fixed.

Line 641: Replace "sub-grid-scale" with "subgrid-scale"

Fixed.

[revised manuscript text omitted]